# Differential coding of reward and movement information in the dorsomedial striatal direct and indirect pathways

Jung Hwan Shin[1], Dohoung Kim [1,2] & Min Whan Jung[1,2,3]

The direct and indirect pathways of the basal ganglia have long been thought to mediate behavioral promotion and inhibition, respectively. However, this classic dichotomous model has been recently challenged. To better understand neural processes underlying reward-based learning and movement control, we recorded from direct (dSPNs) and indirect (iSPNs) pathway spiny projection neurons in the dorsomedial striatum of D1-Cre and D2-Cre mice performing a probabilistic Pavlovian conditioning task. dSPNs tend to increase activity while iSPNs decrease activity as a function of reward value, suggesting the striatum represents value in the relative activity levels of dSPNs versus iSPNs. Lick offset-related activity increase is largely dSPN selective, suggesting dSPN involvement in suppressing ongoing licking behavior. Rapid responses to negative outcome and previous reward-related responses are more frequent among iSPNs than dSPNs, suggesting stronger contributions of iSPNs to outcome-dependent behavioral adjustment. These findings provide new insights into striatal neural circuit operations.

[1] Graduate School of Medical Science and Engineering, Korea Advanced Institute of Science and Technology, Daejeon 34141, Korea. [2] Center for Synaptic Brain Dysfunctions, Institute for Basic Science, Daejeon 34141, Korea. [3] Department of Biological Sciences, Korea Advanced Institute of Science and Technology, Daejeon 34141, Korea. Correspondence and requests for materials should be addressed to M.W.J. (email: mwjung@kaist.ac.kr)

The striatum is known to play crucial roles in diverse behavioral processes such as movement control and reward-based learning[1–4]. Spiny projection neurons (SPNs) of the striatum follow two different output pathways. The direct pathway projects directly to the globus pallidus interna (GPi; in primates) and the substantia nigra pars reticulata (SNr). The indirect pathway projects to the globus pallidus externa (GPe) and then to the SNr/GPi. While direct pathway SPNs (dSPNs) express dopamine D1 receptors, dynorphin, and substance P, indirect pathway SPNs (iSPNs) express dopamine D2 receptors and enkephalin[5,6].

The direct and indirect pathways of the basal ganglia are thought to function in opposition to one another. In the classic rate model of movement control, the direct pathway initiates movement, while the indirect pathway inhibits it[5,7,8]. In support of this theory, stimulation of direct pathway neurons promotes locomotion while stimulation of indirect pathway neurons suppresses it[9–11]. Conversely, ablation of direct pathway neurons reduces locomotion, while ablation of indirect pathway neurons enhances it[10,11]. Arguing against the classic rate model, subsequent studies found concurrent activation of striatal neurons in both the direct and indirect pathways during action initiation[12–15]. These results can be incorporated into a refined classic rate model positing that the direct pathway promotes intended movements, while the indirect pathway inhibits unwanted, competing movements[4,16,17]. Still, there are even more recent findings that are less readily incorporated into models that assume antagonism between the direct and indirect pathways. Not only does unilateral inhibition of both direct and indirect pathway neurons in the dorsolateral striatum induce ipsiversive movements[18], bilateral inhibition of direct and indirect pathway neurons reduces lever pressing behavior and increases action sequence latency[19]. These results suggest a simple dichotomous model assuming antagonism between the two pathways may not do justice to the complexity and subtlety with which the circuits of the basal ganglia control movement.

The direct and indirect pathways are also thought to antagonize one another in reward-based learning, with the direct pathway mediating reinforcement and the indirect pathway mediating punishment and aversion[3,20,21]. Although several studies have documented opposing effects of stimulating or inhibiting the dSPNs and iSPNs on goal-directed behavior[22–28], more recent results again suggest such a dichotomy may be too simplistic[29–31]. Currently, the ways the direct and indirect pathways of the basal ganglia contribute to reward-based learning remain poorly understood. The direct and indirect pathways may mediate behavioral promotion and suppression, respectively, based on different types of signal processing and/or different output connectivity. Here, in an effort to better understand striatal neural processes underlying reward-based learning and movement control, we compared the activity of dSPNs and iSPNs in the dorsomedial striatum in response to positive and negative outcomes and to licking behavior. We found quantitative differences between dSPN and iSPN activity related to value, reward, and licking behavior, which provide new insights into striatal neural circuit operations.

## Results

**Behavior.** To study direct and indirect pathway striatal neural processes underlying reward-based learning and movement control, we trained D1-Cre and D2-Cre mice in a Pavlovian conditioning task. Five D1-Cre and four D2-Cre mice were used in the main physiological experiment. The mice were head-fixed and randomly presented with three distinct odor cues (conditional stimulus, CS; 1 s) paired with three different probabilities (80, 50, and 20%) of reward delivery (5 μl of water) (Fig. 1a, b). Unrewarded trials were paired with a white LED light stimulus (50 lux, 100 ms). The mice showed higher lick rates during the delay period (1 s after CS offset) of higher reward probability trials (one-way ANOVA, $F_{2,318} = 131.5$, $p = 2.4 \times 10^{-42}$; Fig. 1c, d; Supplementary Fig. 1a–j). In the rewarded trials, the mice showed similar lick rates regardless of the odor cue after receipt of the reward (1 s after the first lick following reward delivery: $F_{2,318} = 0.9$, $p = 0.409$; D1-Cre, $F_{2,144} = 1.1$, $p = 0.342$; D2-Cre, $F_{2,171} = 0.1$, $p = 0.888$; Fig. 1d and Supplementary Fig. 1i, j). In unrewarded trials, each odor cue elicited significantly different lick rates even after trial outcomes were revealed (1 s after LED onset; $F_{2, 318} = 62.7$, $p = 1.2 \times 10^{-23}$; D1-Cre, $F_{2,144} = 33.3$, $p = 1.3 \times 10^{-12}$; D2-Cre, $F_{2,171} = 30.1$, $p = 6.1 \times 10^{-12}$; Fig. 1d and Supplementary Fig. 1i, j). These results show that the mice successfully formed cue-reward probability associations.

**Optical tagging of dSPNs and iSPNs.** For optogenetic identification of dSPNs and iSPNs, we injected a double-floxed (DIO) Cre-dependent adeno-associated virus (AAV) vector carrying the gene for channelrhodopsin-2 (ChR2) in-frame with the gene for enhanced yellow fluorescent protein (AAV-DIO-hChR2(H134R)-eYFP, UNC Vector Core) into the left dorsomedial striatum of the five D1-Cre and four D2-Cre mice (Fig. 2a). We then implanted a microdrive array containing one optical fiber and eight tetrodes for unit recording and laser stimulation. We confirmed the localization of the AAV-DIO-hChR2(H134R)-eYFP in the dorsomedial striatum and SNr in D1-Cre mice and in the dorsomedial striatum and GPe in D2-Cre mice via histological examination at the completion of the unit recordings (Fig. 2b, c). We also confirmed proper placement of the optical fiber and tetrodes by histological examination (Fig. 2d).

Total 317 and 413 single units were recorded from D1-Cre (15, 28, 103, 75, and 96 units from five mice) and D2-Cre mice (118, 83, 118, and 94 units from four mice), respectively. We classified the recorded neurons into putative SPNs ($n = 446$, 61.1%), fast-spiking interneurons (FSIs, $n = 130$, 17.8%), and tonically active neurons (TANs, $n = 48$, 6.6%; Supplementary Fig. 2 shows sample activity patterns from the different striatal neuron types). The rest remained unclassified ($n = 106$; 14.5%; Fig. 2). We found 94 units in D1-Cre mice and 102 units in D2-Cre mice that were reliably activated by laser stimulation with short latencies (<6 ms) and low spike jitters (spike latency, D1-Cre mice, 4.2 ± 0.9 ms; D2-Cre mice, 4.3 ± 0.8 ms, mean±SD; SD of spike latency, D1-Cre mice, 0.9 ± 0.5 ms; D2-Cre mice, 1.0 ± 0.5 ms, mean±SD; Fig. 2e, f and Supplementary Fig. 4). The optogenetically confirmed neurons included putative SPNs (77 dSPNs and 75 iSPNs) as well as FSIs (2 from D1-Cre and 4 from D2-Cre mice), TANs (10 from D1-Cre and 15 from D2-Cre mice), and unclassified neurons (5 from D1-Cre and 8 from D2-Cre mice). The proportions of optically tagged neurons were 37.0 and 16.4% of all recorded SPNs and interneurons (FSIs and TANs combined), respectively, in D1-Cre mice, and 31.5 and 18.1%, respectively, in D2-Cre mice. Thus, we obtained roughly similar rates of optogenetic tagging for putative SPNs and interneurons.

A major subclass of FSIs is parvalbumin (PV)-positive neurons[32,33] and TANs are thought to be cholinergic interneurons[33,34]. Because previous studies failed to find D1R and D2R expression in PV neurons[35,36] and D1R expression in cholinergic interneurons[37], we examined ChR2 expressions in PV and cholinergic interneurons using double-immunostaining. We found 6.6% of choline acetyltransferase (ChAT)-positive neurons and 1.2% of PV-positive neurons were co-labeled with ChR2 in D1-Cre mice. We also found 6.2% of ChAT-positive and 2.5% of PV-positive neurons were co-labeled with ChR2 in D2-Cre mice

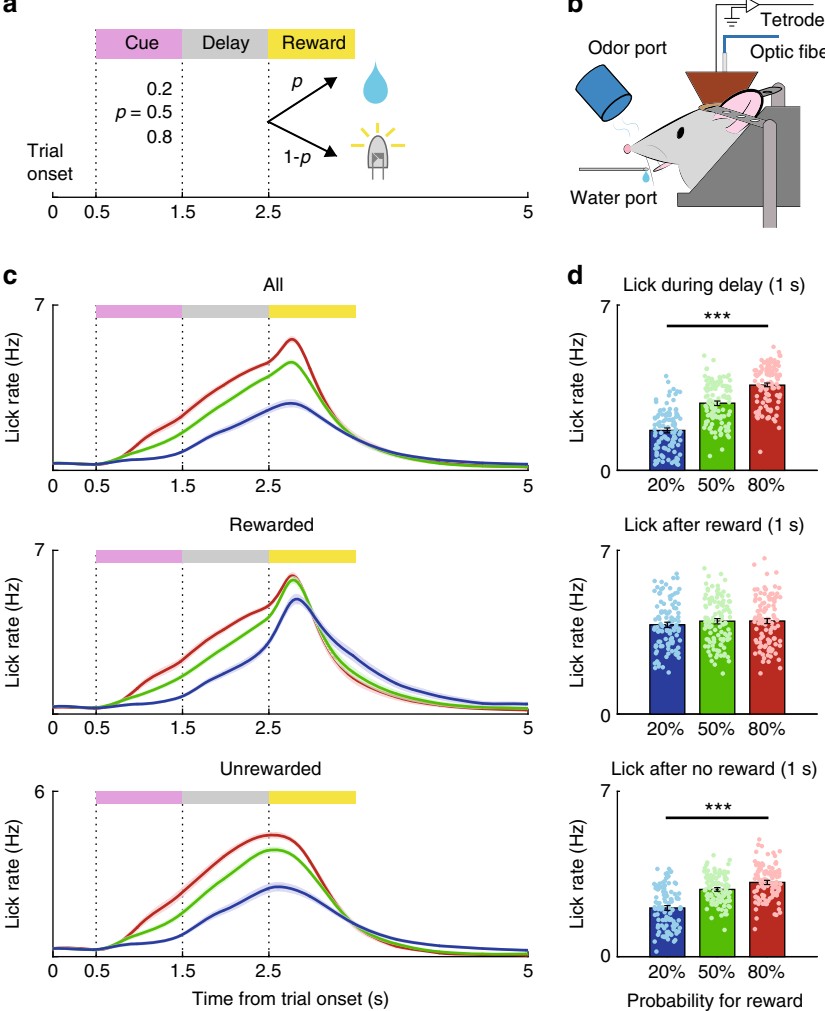

**Fig. 1** Behavioral task. **a** The structure of the probabilistic Pavlovian conditioning task. **b** Schematic of the experimental setting. **c** Licking behavior. Lick density functions ($\sigma = 100$ ms; mean±SEM across 107 sessions performed by five D1-Cre and four D2-Cre mice) are color-coded according to the odor cues that predict different reward probabilities for all trials (top), rewarded trials (middle), and unrewarded trials (bottom). **d** Mean (±SEM) lick rates during the delay period of all trials (top) and during the first 1 s after reward onset in rewarded (middle) and unrewarded (bottom) trials. Reward onset was defined as the time of the first lick after reward delivery in rewarded trials and as the time of LED light onset in unrewarded trials. Circles, individual session data. ***$p < 0.001$ (ANOVA followed by post-hoc Tukey's test for all comparisons)

(Supplementary Fig. 3). These results suggest that our animal lines have both D1R- and D2R-expressing subpopulations of striatal interneurons and/or there were non-specific, cre-independent expressions of ChR2 in the striatal interneurons in our study. We included only putative SPNs in the subsequent analyses (Supplementary Fig. 3 shows the responses of all optogenetically confirmed dSPNs and iSPNs to laser stimulation).

**Neural activity related to reward value**. We first examined dSPN and iSPN responses to reward value using a multiple linear regression analysis (Eq. 1). Many dSPNs and iSPNs respond significantly to reward value (i.e., reward probability) during the cue, delay, and reward periods (Fig. 3a, b). As a control for odor-dependent, rather than value-dependent neuronal firing, we examined the effect of reversing cue-reward probability relationship on value-dependent striatal neuronal activity in a separate group of mice ($n = 3$). A significantly larger population showed similar than reversed activity relationships with value before and after the reversal, indicating that SPN responses to reward value cannot be accounted for by sensory responses (Supplementary Fig. 5a–c). The strength of the value signals in

dSPNs and iSPNs was similar throughout each trial except briefly at cue onset (Fig. 3b). We did find differences between the dSPN and iSPN value signals, however, when we divided the value-responsive neurons into those with positive or negative coefficients (i.e., those whose activity increases or decreases as a function of value, respectively). During the delay and reward periods, we found more SPNs with positive value coefficients among the dSPNs and more SPNs with negative value coefficients among the iSPNs ($\chi^2$-test, positive-coefficient SPNs, delay period, $\chi^2 = 4.8$, $p = 0.028$; first 1 s of the reward period, $\chi^2 = 9.6$, $p = 0.001$; negative coefficient SPNs, delay period, $\chi^2 = 8.1$, $p = 0.001$; first 1 s of the reward period, $\chi^2 = 9.3$, $p = 0.004$; Fig. 3b, c). This shows that although the dSPN and iSPN populations encode value both positively and negatively, there is a bias toward encoding value in opposite directions.

**Neural activity related to reward**. We next looked for evidence that dSPNs and iSPNs also respond to the reward itself (i.e., the trial outcome). Among both dSPNs and iSPNs, we found neurons whose activity increases in response to positive outcomes and neurons whose activity increases in response to negative

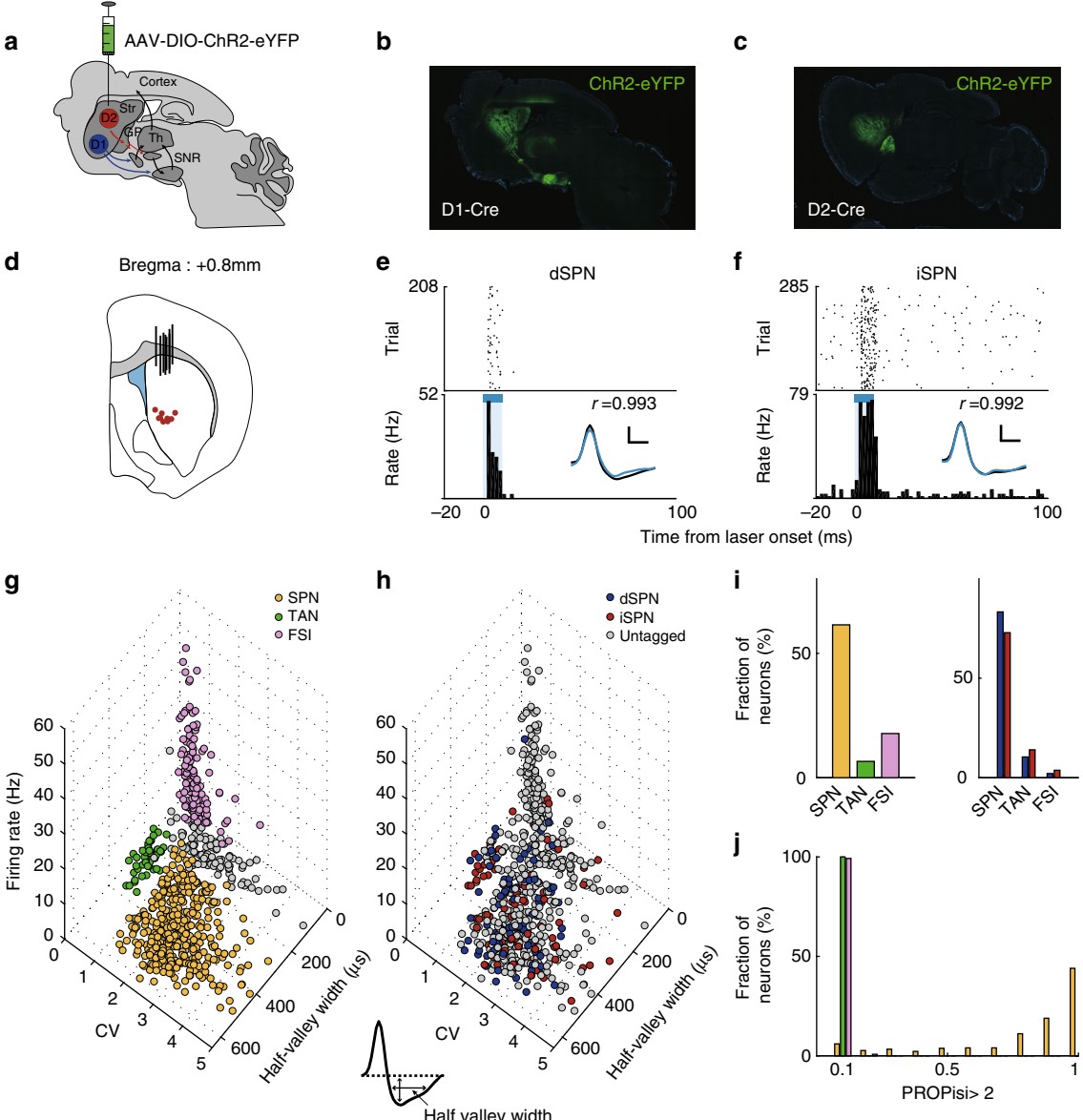

**Fig. 2** Optogenetic identification and classification of D1 and D2 striatal neurons. **a** Schematic of double-floxed cre-dependent AAV viral injection into the dorsomedial striatum shown in a saggital brain section diagram. **b**, **c** Sagittal sections of D1-Cre (**b**) and D2-cre (**c**) mouse brains showing ChR2 expression (green) in the dorsomedial striatum and SNr (D1-cre mouse) or GPe (D2-cre mouse). **d** Stimulation and recording locations shown in a coronal brain section diagram (0.8 mm anterior to bregma). Black vertical bars denote the optical probe locations and red circles indicate the locations of the last recording session for each tetrode, as determined via histological examination. **e**, **f** Examples of optically tagged dSPNs and iSPNs. Top, raster plots. Each row represents one trial and each dot represents a single spike. Bottom, peri-stimulus time histograms (PSTHs). Time 0 denotes laser stimulus onset. The blue bar denotes the 10 ms period of laser stimulation. Inset, averaged spike waveforms of spontaneous (black) and optically driven (blue) spikes. Scale, 250 μs and 30 μV. *r*, correlation between spontaneous and laser-activated spike waveforms. **g** Shown are unit clusters recorded in the main physiological experiment. The recorded units were classified into putative SPNs (yellow), TANs (green), and FSIs (purple) based on their mean discharge rates, the half-valley widths for their averaged spike waveforms, and the CV for their inter-spike intervals. Gray, unclassified units. **h** Optogenetically identified D1 and D2 neurons are indicated in blue and red, respectively. **i** Proportions of SPNs, TANs, and FSIs among all recorded units (*n* = 730; left) and optogenetically identified D1 (*n* = 94; blue) and D2 (*n* = 102; red) neurons (right). **j** Distributions of the different neuron types plotted as a function of the proportion of inter-spike intervals>2 s (PROPisi >2). Colors correspond to those in **g** and **h**

outcomes (Fig. 4a). These reward-related signals began as soon as the trial outcome was revealed (reward onset; aligned to the first lick response following reward delivery in rewarded trials and to light onset in unrewarded trials). During the first 1 s after reward onset, we observed similar fractions of dSPNs and iSPNs responding to reward (36.4 and 37.3%, respectively; $\chi^2$-test, $\chi^2$ = 0.02, $p$ = 0.901; Fig. 4b, c). We next divided these reward-responsive neurons into those with positive and negative reward

coefficients (i.e., neurons with increasing or decreasing activity, respectively, in rewarded trials versus unrewarded trials). Looking at the first 1 s after reward onset, these positive and negative reward neurons both contained similar proportions of dSPNs and iSPNs. Looking only at the first 0.5 s after reward onset, however, a larger fraction of the reward-responsive SPNs with negative coefficients were iSPNs rather than dSPNs ($\chi^2$ = 5.3, $p$ = 0.022; Fig. 4b, c). Additional experiments indicated that neural activity

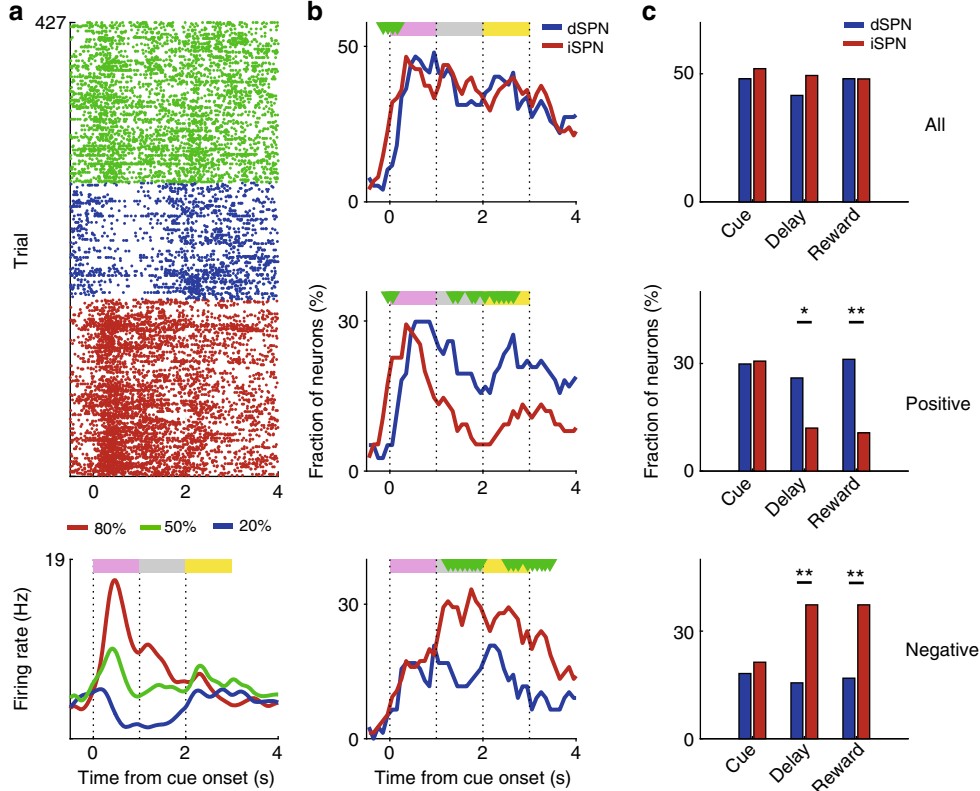

**Fig. 3** dSPN and iSPN responses to value. **a** An example of a value-coding dSPN. A raster plot and spike density function ($\sigma = 100$ ms) are shown. Trials were grouped according to the CS. **b, c** Time course for the proportion of dSPNs ($n = 77$) and iSPNs ($n = 75$) with value-related activity. The analyses included all SPNs (top), SPNs with positive value coefficients (increasing activity with increasing value; middle), or SPNs with negative value coefficients (decreasing activity with increasing value; bottom). **b** Temporal profiles for the proportion of value-coding dSPNs and iSPNs (0.5 s window advanced in 0.1 s steps). Green triangles indicate significant differences between dSPNs and iSPNs ($\chi^2$-test, $p < 0.05$). **c** Fractions of value-coding SPNs in different epochs (1 s each). **$p < 0.01$, *$p < 0.05$ ($\chi^2$-test)

related to negative outcome cannot be accounted for by sensory responses to the light stimulus (Supplementary Fig. 5d–h). Thus, although both dSPN and iSPN populations encode reward both positively and negatively, more iSPNs than dSPNs show negative reward coefficients during the early reward period.

We also examined dSPN and iSPN activity related to previous reward. Consistent with previous reports[38–41], we found slowly-decaying previous reward signals in both dSPN and iSPN populations (Fig. 4d, e). iSPNs, but not dSPNs, show a clear elevation of previous reward signal around the reward onset. In the 1 s window surrounding reward onset, significantly more iSPNs respond to previous reward than dSPNs ($\chi^2$-test, $\chi^2 = 5.5$, $p = 0.016$; Fig. 4e, f). We did not observe any significant difference, however, in the fraction of previous reward-responding iSPNs with positive versus negative coefficients (9.3 and 16.0%, respectively, in the 1 s window surrounding reward onset; $\chi^2$-test, $\chi^2 = 1.5$, $p = 0.219$). These results indicate a more important role of iSPNs in conveying previous reward signals.

To further characterize the reward-related responses of dSPNs and iSPNs, we categorized the response patterns of all the dSPNs and iSPNs based on the difference in firing rates (Δfiring rate) between rewarded and unrewarded trials (Fig. 5a, b). Type 1 neurons are activated in rewarded trials and type 2 neurons are activated in unrewarded trials, both with responses peaking roughly 1 s after reward onset. Type 3 neurons are activated by unrewarded trials and inhibited by rewarded trials, with responses peaking 0.3 s after reward onset. Type 4 neurons show minimal responses in this time window (Fig. 5c). Similar fractions of dSPNs and iSPNs belonged to type 1 and 2 groups ($\chi^2$-test, $\chi^2 =$

0.6 and 0.03, respectively, $p = 0.449$ and 0.854, respectively); however, type 3 consisted of significantly higher fractions of dSPNs than iSPNs ($\chi^2 = 6.5$, $p = 0.010$; Fig. 5d). These results suggest that type 3 iSPNs contribute to the strong negative reward-coding signals in the early reward period (Fig. 4b, c). Consistent with this possibility, we found a significantly larger fraction of iSPN (10, 13%) than dSPN type 3 neurons (three, 4%) during the first 0.5 s of the reward period (Fisher's exact test, $p = 0.045$). Thus, rapid responses to negative outcome seems to be mediated mostly by iSPNs. Most TANs showed typical pause-rebound responses to reward (Supplementary Fig. 6) as reported previously[42].

**Neural activity related to reward prediction error**. We then examined dSPN and iSPN activity related to reward prediction error (RPE). Our analyses so far indicate both dSPN and iSPN populations maintain value signals persistently so that they are combined with reward signals during the reward period. Hence, the signals necessary to compute RPE, namely value and reward signals, were concurrently available for both dSPN and iSPN populations when trial outcome is revealed[38,43]. For the analysis of RPE-related neural activity, we separately examined neural responses to value and reward during the reward period and then examined their relative response directions as in our previous studies[38,40]. Value-dependent firing may be confounded by other factors such as lick rate. To minimize the effect of such potential confounding variables, we examined positive and negative RPE-related neural activity separately. After separating rewarded trials

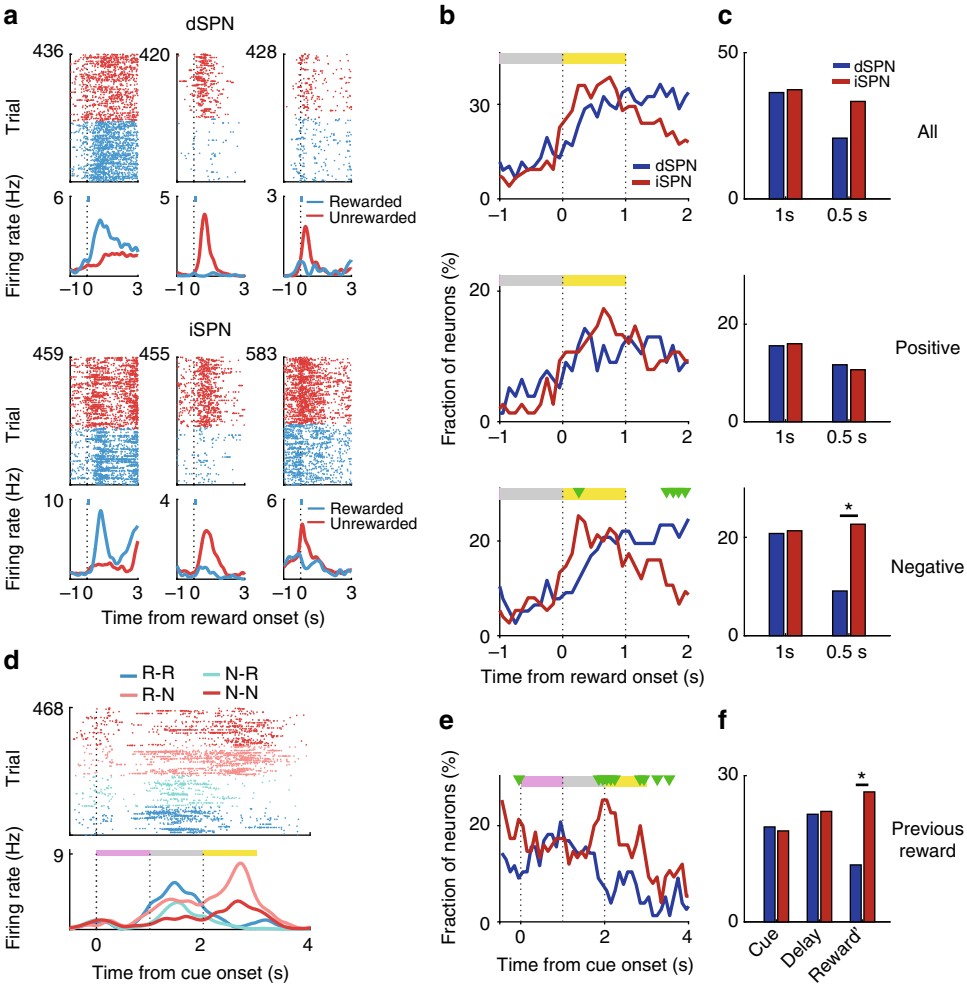

**Fig. 4** dSPN and iSPN responses to reward. **a** Examples of dSPNs and iSPNs responding to reward. Trials were grouped according to trial outcome (rewarded vs. unrewarded). **b, c** Time course of the proportion of dSPNs ($n = 77$) and iSPNs ($n = 75$) with reward-related activity. Analysis included all SPNs (top), SPNs with positive reward coefficients (activity increases in rewarded vs. unrewarded trials; middle), or SPNs with negative reward coefficients (activity decreases in rewarded vs. unrewarded trials; bottom). **b** Temporal profiles for the proportion of reward-coding dSPNs and iSPNs. Graphs follow the same format as those in Fig. 3b. **c** Proportions of reward-coding neurons in different epochs (first 1 or 0.5 s after reward onset). **d** A dSPN responding to current as well as previous rewards. Trials were grouped according to the sequence of the previous and current outcomes (R, rewarded; N, not rewarded). **e, f** Time course for the proportion of dSPNs and iSPNs coding previous reward. Cue, cue period (1 s); Delay, delay period (1 s); Reward', 1 s window centered on reward onset. *$p < 0.05$ ($\chi^2$-test)

from unrewarded trials, we used a multiple linear regression analysis that included lick rate as an explanatory variable (Eq. 2) to examine value-dependent neural activity in the first 1 s after reward onset. Fig 6a separately shows value-dependent firing in rewarded and unrewarded trials of the same neuron shown in Fig. 3a. This neuron fired more during rewarded than unrewarded trials when the trial outcome was revealed. It also showed reduced firing as a function of value in rewarded trials. This neuron, therefore, responded to both the actual outcome (reward) and the predicted outcome (value) in opposite response directions during the reward period. Since RPE is the difference between actual and predicted outcomes[44], we interpreted this as an RPE-coding neuron for rewarded trials (i.e., positive RPE-coding neuron). Note that this neuron increased firing as a function of value before the trial outcome, but decreased firing as a function of value after the trial outcome (Fig. 3a). This cannot be explained by a static maintenance of value-related neural activity and a simple addition of value- and reward-related neural activity. This example shows a dynamic aspect of striatal neural processes underlying value and reward signal processing.

Figure 6b shows regression coefficients for reward (determined using all trials; Eq. 1) and value (determined using either rewarded or unrewarded trials; Eq. 2) for all optogenetically confirmed dSPNs and iSPNs separately for rewarded and unrewarded trials. Consistent with previous findings[38,40], we found SPNs with congruent and incongruent response directions to reward and value. The neurons with incongruent response directions can be considered as RPE-coding neurons, because RPE is the difference between actual and expected rewards[44]. Of the 77 dSPNs and 75 iSPNs, 13 dSPNs (16.9%) and 9 iSPNs (12.0%) responded significantly to both reward (all trials) and value in rewarded trials, and 12 dSPNs (15.6%) and 12 iSPNs (16.0%) responded significantly to both reward (all trials) and value in unrewarded trials. Of the SPNs significantly responsive to both reward and value, eight dSPNs and seven iSPNs in the rewarded trials and six dSPNs and seven iSPNs in the unrewarded trials showed responses consistent with a role in RPE-coding (Fig. 6b, c). We did not find a significant difference between the fractions of positive and negative RPE-coding neurons among the dSPN ($\chi^2$-test, $\chi^2 = 0.3$, $p = 0.575$) or iSPN population ($\chi^2 = 0$, $p = $

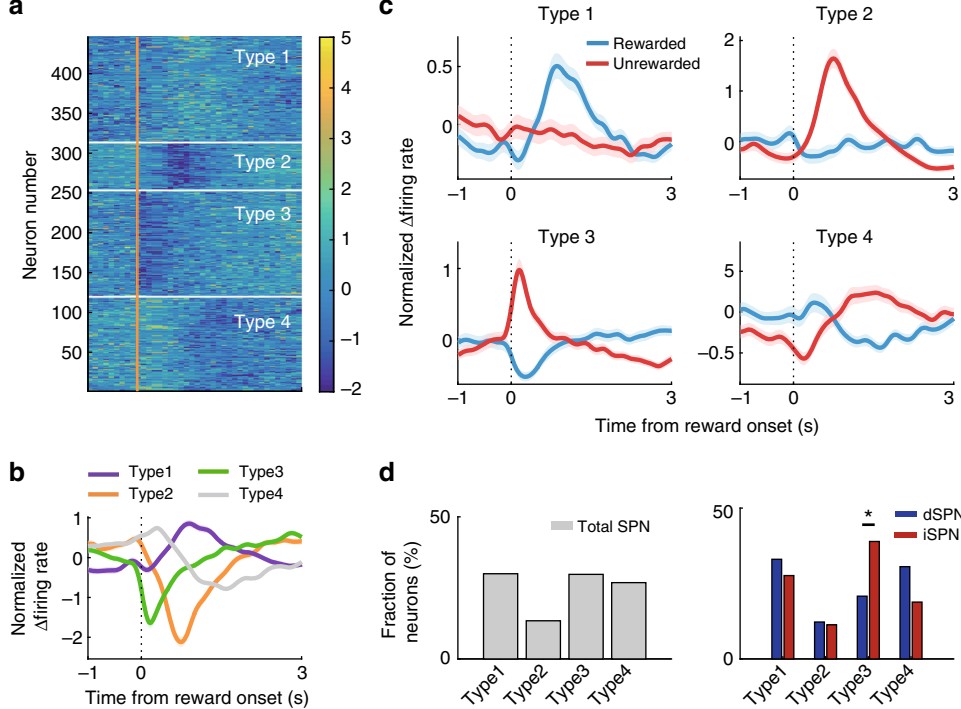

**Fig. 5** Types of responses to reward. **a** The responses of all dSPNs and iSPNs to reward are shown as z-normalized Δfiring rate. Neural responses around the time of reward onset (1 s before and 3 s after) were converted as the difference in firing rates (Δfiring rate) between rewarded and unrewarded trials in 100 ms time bins, and then normalized to unit variance based on the mean and SD. **b** The z-normalized Δfiring rates were classified into four groups using k-means clustering along the first three components of a principal components analysis. **c** Mean normalized responses of each type plotted separately for rewarded vs. unrewarded trials. Shading, SEM. **d** Fractions of total SPNs (optically tagged and untagged SPNs combined, $n = 446$), dSPNs ($n = 77$), and iSPNs ($n = 75$) in types 1–4. *$p < 0.05$ ($\chi^2$-test)

1.000; Fig. 6d). Nor did we find a significant difference between the fractions of positive RPE-coding dSPNs and iSPNs ($\chi^2 = 0.05$, $p = 0.827$) or between negative RPE-coding dSPNs and iSPNs ($\chi^2 = 0.1$, $p = 0.734$). In addition, we found no significant deviation from an equal distribution in the proportions of dSPNs and iSPNs whose activity increases or decreases as a function of RPE (Fisher's exact test, $p = 0.367$ and 0.528, respectively). Thus, not only are dSPNs and iSPNs similarly responsive to positive and negative RPE, they also represent RPE in both the positive and negative directions.

**Neural activity related to licking behavior.** We next examined SPN activity related to lick onset (licks occurring >2 s after the previous lick) and lick offset (licks occurring >2 s before the following lick; examples in Fig. 7a, b)[45,46]. Most instances of lick onset occurred before delay offset, while most instances of lick offset occurred after delay offset (Supplementary Fig. 1). We aligned all dSPN and iSPN responses (z-normalized according to the mean and SD of neural activity in 100 ms bins) that occurred between 1 s before and 3 s after lick onset/offset according to their peak responses (Fig. 7c). Consistent with previous reports on locomotion- and lever press-related SPN activity[12,13,15], we found simultaneous activation of dSPNs and iSPNs at lick onset, with peak responses of some dSPNs and iSPNs preceding lick onset (Fig. 7a, c and Supplementary Fig. 7). We also found simultaneous activation of dSPNs and iSPNs at lick offset (Fig. 7d and Supplementary Fig. 7). Many dSPNs and iSPNs responded to multiple variables across multiple stages (Supplementary Fig. 8). Thus, consistent with previous findings[12,13], we found concurrent activation of both dSPNs and iSPNs in association with the initiation and termination of a lick bout.

For a more nuanced analysis of lick offset-related neural responses, we separated lick bouts according to their durations to avoid potential overlap between lick offset-related neural responses and lick onset-related neural responses in short-duration lick bouts. This analysis revealed that dSPNs are more strongly activated at lick offset (i.e., following the last lick of a lick bout) than iSPNs, especially for the longer lick bouts (Fig. 7e, f). Note the largely selective enhancement of dSPN responses in association with the offset of a relatively long lick bout (Fig. 7f, lick bouts >1.5 and >2 s). Consistent with these results, we found in a multiple regression analysis[45,46] that lick offset coefficients for dSPNs are more positive than those of iSPNs (Supplementary Fig. 9). Thus, unexpectedly, we found stronger activation of dSPNs than iSPNs during lick offset.

To directly test a role for dSPNs in suppressing ongoing licking behavior, we examined the effects of bilateral stimulation of the direct pathway dorsomedial striatal neurons in a separate group of D1-Cre mice ($n = 3$; Fig. 7g; ChR2 expressions and optical probe locations confirmed by histological examination as in Fig. 2). A continuous laser pulse for 1 s (473 nm) in the same experimental setting (head-fixed condition) strongly suppressed ongoing licking behavior, but licking resumed immediately after laser stimulus offset (Fig. 7h–j). Because optogenetic stimulation of D1 striatal neurons increased movement velocity in an earlier study[9], we also tested these three mice under a freely moving condition in a square box ($30 \times 30 \times 30$ cm). For this, as in the previous study[9], we delivered a continuous bilateral stimulus (473 nm) for 30 s with 30-s inter-trial intervals. Here, too, we observed a laser-stimulation-induced increase in movement velocity (Fig. 7k, l). Thus, dSPN stimulation suppressed ongoing licking behavior, while increasing movement velocity of freely moving mice.

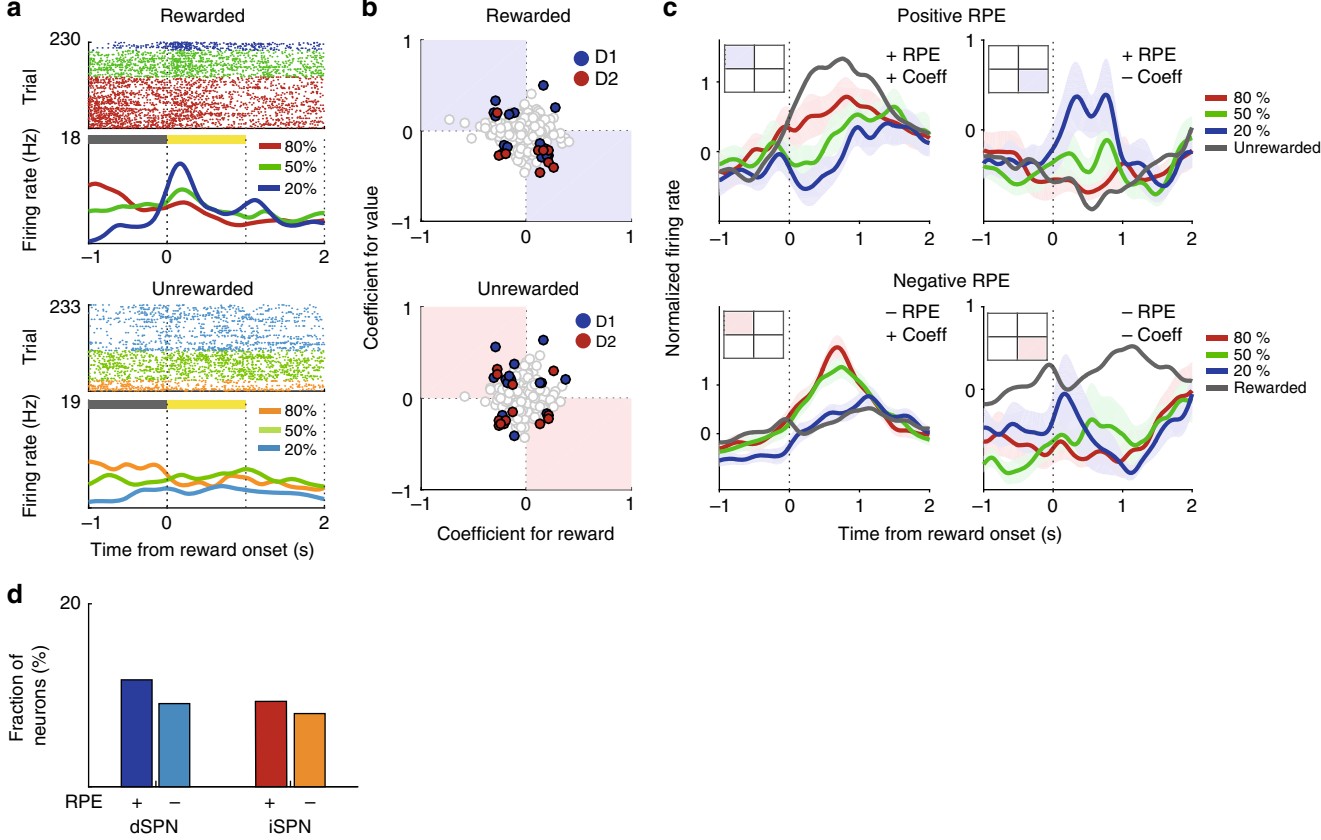

**Fig. 6** dSPN and iSPN responses to RPE. **a** An example of a positive RPE-coding neuron. This is the same neuron shown in Fig. 3a, but its responses in rewarded vs. unrewarded trials are shown separately. **b** The scatter plots show regression coefficients for reward (Eq. 1; all trials) and value in rewarded or unrewarded trials (Eq. 2). Blue and red circles indicate dSPNs ($n = 13$ and 12 for rewarded and unrewarded, respectively) and iSPNs ($n = 9$ and 12 for rewarded and unrewarded, respectively), respectively, that were significantly responsive to both reward and value. Those in the second and fourth quadrants (i.e., those with opposite responses to reward and value) were considered RPE-coding SPNs. **c** Mean normalized firing rates (spike density functions, $\sigma = 100$ ms) of RPE-coding SPNs (from all SPNs, $n = 34$ and 49 for rewarded and unrewarded, respectively) in each quadrant. Inset squares indicate the quadrant in **b** being analyzed. Shading, SEM. Mean normalized firing rates in unrewarded (top) and rewarded (bottom) trials are shown in gray for comparison. **d** Fractions of positive (+) and negative (−) RPE-coding SPNs

Finally, we asked whether there is any correlation between dSPN and iSPN discharges and lick rate (Fig. 8a, b). For this, we calculated mean discharge rates and mean lick rates for all lick bouts for each SPN, and calculated the Pearson's correlation coefficient between them. We found a significant correlation ($p < 0.01$) between the activity and lick rate for 24.6% of dSPNs and 32.0% of iSPNs (Fig. 8c; no significant difference between them; $\chi^2$-test, $\chi^2 = 1.0$, $p = 0.316$). When we included all optically tagged SPNs in the analysis, both dSPNs and iSPNs had mean correlation coefficients significantly smaller than zero ($-0.031$ and $-0.047$, respectively; $t$-test, $t_{76} = -2.1$ and $-2.7$, respectively $p = 0.044$ and 0.009, respectively). These, too, were not significantly different from one another (Wilcoxon's rank-sum test, $p = 0.470$). Seven (36.8%) of the dSPNs whose activity showed a significant correlation with lick rate had positive correlation coefficients and 12 (63.2%) had negative correlation coefficients. This does not differ significantly from an equal distribution ($\chi^2$-test, $\chi^2 = 1.5$, $p = 0.220$). Of the iSPNs with significant correlations, 6 (25.0%) had positive coefficients and 18 (75.0%) had negative coefficients. This distribution *does* represent a significant deviation from an equal distribution ($\chi^2 = 7.1$, $p = 0.007$; Fig. 8d, e). Still, in the comparison between dSPNs and iSPNs, the distribution of neurons with positive and negative coefficients did not differ significantly from an equal distribution ($\chi^2 = 0.06$ and 1.7, respectively, $p = 0.810$ and 0.193, respectively). In summary, we

found both positive and negative correlations between dSPN and iSPN firing and lick rate, but the correlations were slightly but significantly negative especially for iSPNs.

## Discussion
Here, we examined the firing patterns of optogenetically identified dSPNs and iSPNs in the dorsomedial striatum during a probabilistic Pavlovian conditioning task. We found both dSPN and iSPN populations are responsive to a diverse array of reward- and tongue movement-related variables; they are responsive to value, reward, and RPE as well as to the initiation and termination of licking behavior. Both dSPN and iSPN populations concurrently represent opposing signals (positive and negative outcomes, positive and negative RPEs, and lick initiation and termination), and they show activity-increasing as well as -decreasing responses to each variable. There were, however, clear and quantifiable differences between the activities of the dSPN and iSPN populations. While dSPNs tend to increase firing with increasing value, iSPNs tend to reduce firing with increasing value. More iSPNs convey rapid negative outcome and previous outcome signals than dSPNs. dSPNs are activated more strongly than iSPNs by lick offset.

We found examples of dSPNs and iSPNs whose activity increases as a function of value, but also examples whose activity decreases. There is a clear bias, however, among dSPNs

toward activity increases and among iSPNs toward activity decreases. This suggests the dorsomedial striatum may signal estimated value through the relative activity levels of its direct and indirect pathways. Such a scheme is consistent with previous manipulation studies that found opposing effects on goal-directed behavior of selective stimulation or inhibition of direct vs. indirect pathway neurons[22–28]. A more recent study

reported that the stimulation of direct and indirect pathway neurons induces opposing responses in the downstream brain areas[47]. The relative activity of the direct and indirect pathways, along with their distinct output connectivity, may determine the likelihood with which an animal will choose a particular target.

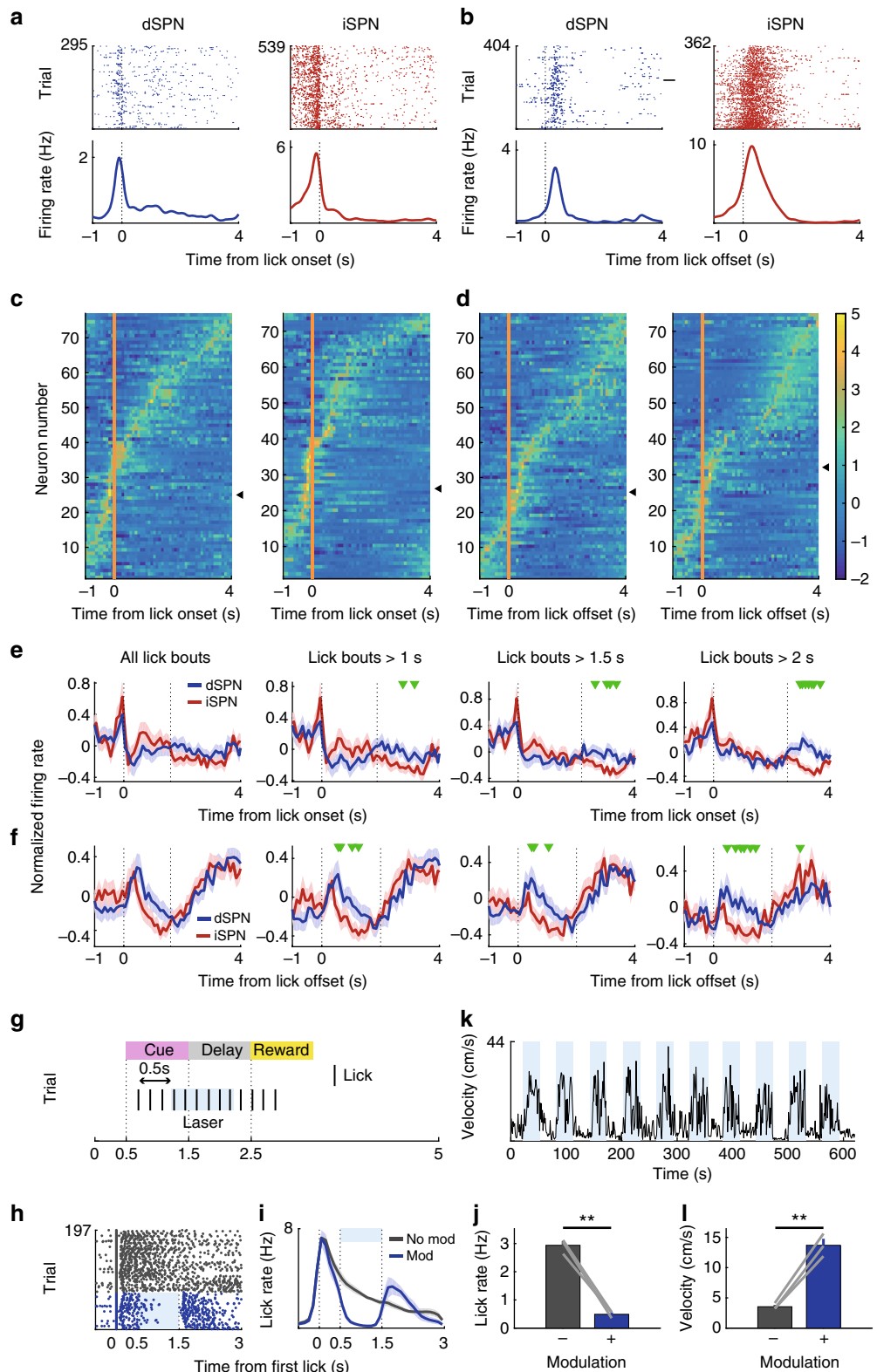

We found that both dSPNs and iSPNs respond to positive and negative outcomes as well as to positive and negative RPE. Our results indicate that the direct and indirect pathways process opposing reward signals together rather than each processing a single type of reward signal. There are, however, quantitative differences. More iSPNs than dSPNs are rapidly activated by negative outcomes and inhibited by positive outcomes (i.e., type 3 neurons; Fig. 5). Also, iSPNs carry stronger previous reward signals (a larger fraction of previous reward-responsive neurons) around reward onset than dSPNs, so that iSPNs carry current and previous reward signals simultaneously. These results suggest a more important role of iSPNs in outcome-dependent adjustment of behavior than dSPNs. Behavioral choices in humans and animals are controlled by multiple underlying processes. The fast responses of iSPNs may contribute to rapid, negative outcome-dependent behavioral adjustments (e.g., lose-switch), whereas their previous outcome signals may contribute to adjusting behavior according to the history of past outcomes (e.g., reinforcement learning) (c.f. [39,48–51]). There exists a large body of evidence implicating D2 receptors in reversal learning[52]. We previously showed D2 receptors are more important than D1 receptors in optimizing choice behavior in dynamic, uncertain environments; D2, but not D1, receptor-knockout mice have difficulty updating action value in response to positive and negative outcomes in a dynamic foraging task[48]. A recent study in monkeys reported injection of a D2, but not D1, antagonist into the dorsal striatum impairs learning from past outcomes[53]. These results suggest a more important role of the indirect pathway in outcome-dependent adjustment of behavior. It would be informative to measure and manipulate the reward-dependent responses of dSPNs and iSPNs in a free-choice task.

The direct and indirect pathways are thought to antagonize one another by facilitating and suppressing movement, respectively[4,5,7,8,16,17]. Here, we found dSPNs are more strongly activated than iSPNs at lick offset. Our results are difficult to reconcile with the conventional model that assumes a prokinetic role for the direct pathway and an antikinetic role for the indirect pathway. In a recent study[19], strong stimulation of direct pathway neurons in the dorsolateral striatum suppressed, rather than facilitated, lever presses. Moreover, lever presses resumed immediately upon stimulation termination and the total number of lever presses was not reduced by the stimulation, suggesting that direct pathway neurons momentarily suppress ongoing lever presses while being stimulated. Although strong stimulation of indirect pathway neurons also suppressed lever presses, immediate resumption of lever presses was not observed upon stimulation termination. Consistent with these results, we found strong stimulation of direct pathway neurons suppresses licking, but this licking resumes upon stimulation

termination. We also replicated the previous finding that continuous stimulation of direct pathway neurons in the dorsomedial striatum increases movement velocity in freely moving mice[9]. Thus, direct pathway stimulation suppresses ongoing responses (licking in the present study and lever pressing in ref. [19]) while increasing movement velocity in freely moving mice. It is not straightforward to link specific behavior-related dSPN activity to stimulation effect because optogenetic stimulation presumably activates a general population of dSPNs that are related to diverse behaviors. Given that striatal neurons seem to be composed of functionally distinct spatial clusters[15,54], modulating specific functional dSPN or iSPN clusters should help reveal the precise mechanisms by which the direct and indirect pathways control movement.

Some studies suggest the absence of D1R and D2R expression in PV neurons[35,36] and the absence of D1R expression in cholinergic interneurons[37]. However, our double-immunostaining indicates the presence of ChR2-expressing PV neurons and TANs in both D1-Cre and D2-Cre mice. There are also studies reporting that striatal cholinergic interneurons express D1R mRNA, albeit at a lower level (20-25%) compared with D2R mRNA[55,56], and a small fraction of PV interneurons and somatostatin-expressing striatal interneurons express D2R mRNA[57] and D1R mRNA[58], respectively. These results explain why we obtained optogenetically tagged FSIs and TANs in both D1-Cre and D2-Cre mice. Nevertheless, we cannot rule out the possibility of non-specific, cre-independent expressions of ChR2 in FSIs and TANs in our mice, which may raise a concern regarding the validity of optogenetic tagging. Our conclusions on dSPNs and iSPNs are likely to be valid, however, for the following reasons. First, histological examinations revealed a clear segregation between dSPN and iSPN populations (Fig. 2). Second, our unit classification was sufficiently stringent. We included only those units with L-ratio <0.1 and isolation distance >19[59], and those neurons that were not clearly segregated were left as unclassified and excluded from the analysis. Furthermore, we obtained similar results when we analyzed only those SPN unit clusters with L-ratio <0.05, which are very well-isolated unit clusters (72 dSPNs and 65 iSPNs) and when we excluded relatively high-rate SPNs (cut-off value, one SD below mean firing rate of TANs, 3.61 Hz; 68 dSPNs and 68 iSPNs were analyzed). Thus, the chance for interneurons to be erroneously classified as SPNs is low. Third, we used reasonably stringent criteria for optogenetic tagging (activation window, 6 ms; log-rank test, $p < 0.01$; spike waveform correlation >0.8) and obtained similar results when we applied more stringent criteria (log-rank test, $p < 0.001$; spike waveform correlations >0.95). Fourth, lick offset- and previous reward-related activity increases were largely selective to dSPNs and iSPNs,

**Fig. 7** dSPN and iSPN responses to licking. **a**, **b** Examples of dSPNs and iSPNs responding to lick onset and offset. Time 0 denotes the time of the first (**a**) or last (**b**) lick in a lick bout. **c**, **d** Normalized responses ($z$-scores) of all dSPNs ($n = 77$) and iSPNs ($n = 75$) to lick onset and offset (100 ms bins). Triangles indicate example neurons from **a** and **b**. **e**, **f** Mean normalized responses of dSPNs and iSPNs to lick onset (**e**) and offset (**f**) are shown for all lick bouts and for lick bouts with >1, 1.5, or 2 s durations. The second vertical dashed line denotes the mean lick offset time (**e**) or mean next lick onset time (**f**). Green triangles denote significant differences between dSPNs and iSPNs ($t$-test, $p < 0.05$). **g–l** The effect of bilateral stimulation of D1 striatal neurons on licking and movement velocity. **g** Laser stimulation schematic for head-fixed mice. A continuous laser pulse (1-s duration) was given 0.5 s after the first lick post-cue onset in a randomly chosen subset (30%) of trials. **h** An example session showing the effect of laser stimulation on licking. Each row is one trial and each tick mark denotes a lick. Trials were grouped according to laser stimulation. **i** Group data (three D1-cre mice; each tested in two sessions). Lick density functions ($\sigma = 10$ ms) with and without the laser stimulus. **j** Mean lick rates with and without the laser stimulus (1-s period beginning 0.5 s after the first lick). **k** The same mice were also tested in a free-movement condition. Laser stimulation was applied continuously for 30 s with 30-s inter-trial intervals. A sample movement velocity trace during 10 laser-on–laser-off episodes (blue indicates laser-on). **l** Mean movement velocity under laser-on and laser-off conditions. **p < 0.01, paired $t$-test

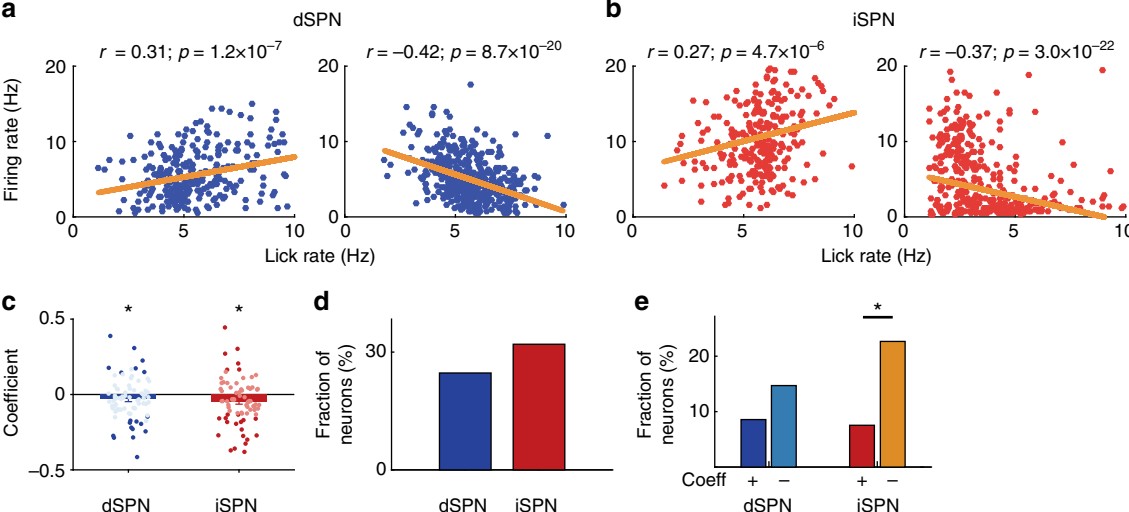

**Fig. 8** dSPN and iSPN responses to lick rate. **a**, **b** Examples of dSPNs (**a**) and iSPNs (**b**) whose firing rates show significant correlation with lick rate. Each dot represents mean lick rate and mean discharge rate during a lick bout. *r*, Pearson's correlation coefficient. **c** Distributions of correlation coefficients between lick rate and firing rate (77 dSPNs and 75 iSPNs). Bold colors denote neurons whose activity shows significant correlation with lick rate. *significantly different from 0 (*t*-test, $p < 0.05$). **d** Fractions of dSPNs and iSPNs whose activity shows a significant correlation with lick rate. **e** Fractions of dSPNs and iSPNs whose activity shows a significant positive (+) or negative (−) correlation with lick rate. *$p < 0.05$ ($\chi^2$-test)

respectively. These results argue against cross-contamination between dSPN and iSPN populations

The roles the direct and indirect pathways play in reward-based learning and motor control may vary across the different areas of the striatum. The dorsomedial, dorsolateral, and ventral striatum seem to act in separate cortico-striatal loops that serve different aspects of behavioral control[2,6,60,61]. With regard to reward-based learning, previous manipulation studies in the dorsomedial striatum yielded results supporting the view the direct and indirect pathways mediate reinforcement and punishment/aversion, respectively[24,25,27]. By contrast, stimulation of dSPNs and iSPNs in the dorsolateral striatum enhanced behavioral responses, albeit in different ways[31]. For the ventral striatum, where the extent of D1 and D2 receptor segregation remains controversial[62], there are findings that are consistent[11,22,23,26] as well as inconsistent[63,64] with the model assuming opposing roles of the direct and indirect pathways in reward-based learning. Thus, further studies are required to determine whether and how the direct and indirect pathways contribute to reward-based learning across the different loops of the striatum.

With respect to motor control, optogenetic modulation of direct and indirect pathway neurons in the dorsomedial striatum produced results supporting a classic rate model[9]. Later studies in the dorsal[12,13] and dorsolateral striatum[19], however, argue against such a model. Our results, obtained in the dorsomedial striatum, also argue against the classic model. Licking-related neural responses have been recorded in the dorsolateral striatum[65], and a stimulation study identified an area in the dorsolateral striatum that induces licking behavior[66]. It is unclear whether the dorsomedial striatum contains a similar area that controls licking behavior. While the dorsolateral striatum receives direct projections from the mouth/tongue areas of the sensory/motor cortices, the dorsomedial striatum does not[67]. Nevertheless, we found a significant correlation between neural activity in the dorsomedial striatum and the onset, offset, and rate of licking behavior. It has been proposed that the ventral, dorsomedial, and dorsolateral striatum are in charge of coarse, intermediate, and fine control of behavior, respectively[2]. According to this hypothesis, our results may be related to dorsomedial striatal contributions to the intermediate level of behavioral control. The direct pathway of

the dorsomedial striatum may play a role in stopping ongoing behavior and initiating another behavior, whereas finer control of specific behavior may be controlled by the dorsolateral striatum, which remains to be determined. So far, few studies have explored the role the direct and indirect pathways in the dorsomedial and ventral striatum have on animal movement[9]. Further studies measuring and manipulating the activity of the direct and indirect pathways in the dorsomedial, dorsolateral, and ventral striatum are necessary to quantify the contributions the different cortico-striatal loops make to motor control.

## Methods

**Animals**. C57BL/6J BAC transgenic mouse lines expressing Cre recombinase under control of the dopamine D1 and D2 receptors (EY217 and ER43, respectively) were obtained from Gene Expression Nervous System Atlas. The mice were deprived of water and their body weights were maintained at >80% ad libitum levels throughout the experiments. They were individually housed, and all experiments were conducted in the dark phase of a 12 h light/dark cycle. Only male (D1-Cre, $n = 10$; D2-Cre $n = 5$) mice were used. Five D1-Cre and four D2-Cre mice were used for the main physiological experiments, two D1-Cre and one D2-Cre mice were used for the physiological experiments examining neural responses to sensory cues (Supplementary Fig. 5), and three D1-Cre mice were used for the behavioral experiment examining laser stimulation effects on licking and movement velocity (Fig. 7g–l). All mice were 12–18 weeks old at the time of physiological recording. The experimental protocol was approved by the Animal Care and Use Committee of the Korea Advanced Institute of Science and Technology (Daejeon, Korea).

**Behavioral task**. The mice were trained in a probabilistic Pavlovian conditioning task (Fig. 1). Each mouse's head was fixed to the recording apparatus using a custom-designed metal plate. Half a second after trial onset (signaled by a clicking solenoid valve), a 1 s pulse of one of three odors (citral, isoamyl acetate, and (−) carvone diluted 1/1000 v/v in mineral oil) was delivered with a custom-designed olfactometer[46] in a pseudorandom order. No odor cue was presented more than three times in a row. After a 1 s delay, either a reward (5 μl of water) or a non-reward (a 100 ms LED light pulse at 50 lux) was delivered with a predetermined probability (20, 50, or 80%). The probability of each odor–reward combination varied across animals. The duration of each inter-trial interval was determined randomly between 4 and 5 s. Licking responses were detected with an infrared photobeam sensor placed in front of the water port. Before beginning unit recordings, the mice were trained in the task until their delay-period lick rates differed significantly according to the expected reward probability (one-way ANOVA followed by post-hoc Tukey's tests, $p < 0.05$ for all comparisons) (4.5 ± 1.6 d, mean ± SD). The mice performed 360–480 trials per daily recording session. Rewarded trials in which there was no licking response between the delay offset

and the onset of the next trial were considered incomplete. Trials following such incomplete trials were excluded from the analysis because these trials always provided a reward.

**Virus injection**. The mice were anesthetized with isoflurane (1.5–2.0% [v/v] in 100% oxygen), and a small burr hole (diameter, 0.5 mm) was made 0.8 mm anterior and 1.4 mm lateral to the bregma to target the dorsomedial striatum unilaterally (left hemisphere; physiological experiments) or bilaterally (behavioral experiment). A bolus of 1 μl of virus (AAV5-EF1a-DIO-ChR2(H134R)-eYFP, UNC Vector Core) was injected 2.8 mm below the brain surface at a rate of 0.1 μl/min. The injection needle was then held in place for 5 min after the injection.

**Neurophysiology and optogenetics**. For those mice used in the physiological experiments, a microdrive array containing an optic fiber (core diameter, 200 μm) and eight tetrodes was implanted in the left dorsomedial striatum (0.8 mm anterior and 1.4 mm lateral to the bregma; 2.3 mm ventral to the brain surface) immediately after virus injection. The optic fiber was held in place throughout the experiment, but the tetrodes were advanced 50–100 μm per day once unit recording began. Unit signals were amplified 10,000×, band-pass filtered between 600 and 6000 Hz, digitized at 32 kHz, and stored on a personal computer using the Cheetah data acquisition system (Neuralynx). Laser pulses (473 nm; 10 ms; Doric corp/Omicron Phoxx) were delivered at 1 Hz with variable intensities (0.5–1.5 mW at optic fiber tip; 120–300 pulses) at the end of each recording session to identify laser-activated neurons. A cathodal electrolytic current (20 s, 30 μA) was applied through one channel of each tetrode at the end of the final recording session to leave marking lesions. Coronal and sagittal sections (40 μm thick) of the brain were prepared according to a standard histological procedure[68], and the brain sections were examined under a light microscope to locate electrode tracks and electrolytic lesions. A slide scanner microscope (Zeiss Axioscan) was used to verify ChR2 expression.

For the D1-Cre mice ($n = 3$) used in the behavioral experiments, only optic fibers were bilaterally implanted. Two weeks after viral injection and optic fiber implantation, the mice were placed in a video-monitored square chamber (30 × 30 × 30 cm). After a 10-min habituation period, laser stimulation was delivered in a series of 10 trials. Each trial consisted of continuous laser illumination (5 mW at optic fiber tip, 473 nm, 30 s) followed by a 30-s laser-off period. Nose position and body center were tracked using ETHOVISION XT 11.5 (Noldus). The same mice were then tested in the probabilistic Pavlovian conditioning task under head-fixed conditions. Only one odor cue associated with a 90% reward probability was used in this experiment. Continuous laser illumination (5 mW at fiber tip, 473 nm, 1 s) was delivered 0.5 s after the first lick of each trial in a randomly chosen subset of trials amounting to 30% of the total trials.

**Unit isolation and classification**. Putative single units were isolated offline by manual cluster cutting of various spike waveform parameters using the MClust software (A.D. Redish). Only those clusters with L-ratio <0.1 (0.016±0.020, mean ±SD, $n = 730$) and isolation distance >19 (65.9±94.0)[59] were included in the analysis. Mean (±SD) peak amplitude of the recorded units was 160.6 ± 78.5 μV (noise band amplitude, 30–35 μV). Mean (±SD) isolation distance, L-ratio, and spike amplitude were 0.018 ± 0.018, 46.7 ± 33.2, and 143.7 ± 51.5 for dSPN ($n = $ 77), and 0.025 ± 0.025, 41.1 ± 25.6, and 149.1 ± 79.2 for iSPN ($n = 75$) (Supplementary Fig. 2). The recorded units were classified into putative SPNs, FSIs, and TANs based on mean discharge rates, coefficients of variation (CV) of their inter-spike intervals, and half-valley widths of the filtered spike waveforms (Supplementary Fig. 2). The mean (±SD) firing rate, the CV of the inter-spike intervals, and the half-valley width were 1.2 ± 1.8 Hz, 2.57 ± 1.53, and 397.5 ± 91.8 μs, respectively, for the putative SPNs, 22.4 ± 9.8 Hz, 1.50 ± 2.70, and 124.8 ± 18.2 μs, respectively, for the putative FSIs, and 5.8 ± 2.2 Hz, 0.68 ± 0.15, and 296.5 ± 48.5 μs, respectively, for the putative TANs. Only putative SPNs were included in the analysis.

**Identification of laser-responsive neurons**. To be qualified as laser-activated, neurons had to meet two criteria[46,69]. First, the latency to the first spike during the 6 ms window after laser stimulation onset should be significantly (log-rank test, $p$ < 0.01) lower than the latency to the first spike in a similar 6 ms window in the absence of laser stimulation. Second, the correlations between laser-driven and spontaneous spike waveforms should be >0.85. Increasing the stringency of these criteria (i.e., log-rank test, $p < 0.001$; spike waveform correlation >0.95) reduced the number of laser-responsive SPNs (dSPNs, from 77 to 64; iSPNs, from 75 and 72), but yielded similar results.

**Regression analysis**. Neural activity related to value, reward, and lick rate was examined using the following regression model:

$$F(t) = a_0 + a_1 \cdot O(t) + a_2 \cdot V(t) + a_3 \cdot L(t) + a_4 \cdot O(t-1) + a_5 \cdot V(t-1),$$
(1)

where $F(t)$ represents neural firing rate in a given analysis window, $O(t)$, $V(t)$, and $L(t)$ are reward (i.e., trial outcome; 1 if rewarded, 0 if unrewarded), value (reward

probability; 0.2, 0.5, or 0.8), and lick rate, respectively, in each analysis window in trial $t$, and $a_0$–$a_5$ are regression coefficients. To analyze positive and negative RPEs, we divided the trials into rewarded and unrewarded, and examined neural activity during the reward period with the following regression model:

$$F(t) = a_0 + a_1 \cdot V(t) + a_2 \cdot L(t) + a_3 \cdot O(t-1) + a_4 \cdot V(t-1).$$
(2)

The following regression model was used to separately examine neural activity related to the onset, maintenance, and offset of licking behavior[45,46],

$$\log S(t) = \beta_0 + \sum_{n=-2}^{2} \beta_n^{ts} F_{t-n}^{ts} + \sum_{n=-2}^{17} \beta_n^{A} F_{t-n}^{A} + \sum_{n=-2}^{17} \beta_n^{B} F_{t-n}^{B} + \sum_{n=-2}^{17} \beta_n^{C} F_{t-n}^{C} +$$
$$\sum_{n=-2}^{22} \beta_n^{rw} F_{t-n}^{rw} + \sum_{n=-2}^{22} \beta_n^{non-rw} F_{t-n}^{non-rw} + \sum_{n=-6}^{14} \beta_n^{lk} F_{t-n}^{lk} + \sum_{n=-6}^{29} \beta_n^{lon} F_{t-n}^{lon} + \sum_{n=-6}^{79} \beta_n^{loff} F_{t-n}^{loff}$$
(3)

where the meaning of each variable is as follows: $S(t)$, z-scored mean firing rate during each 100 ms bin; $\beta_0$–$\beta_n^{loff}$, regression coefficients; and $F_{t-n}$, occurrence of task event at time $t-n$ (1 if event occurred, 0 otherwise). The superscripted symbols mean the following: ts, trial onset; A–C, odor cues; rw, reward; non-rw, no-reward; lk, mid-bout licks; lon, lick onset; and loff, lick offset. Lick onset was defined as a lick that occurred >2 s since the previous lick, and lick offset as a lick when the interval until the next lick was >2 s. All other licks with shorter inter-lick intervals were defined as mid-bout licks.

**Statistical analysis**. The statistical significance of the regression coefficients was determined with $t$-tests. Significant differences between fractions of dSPNs and iSPNs were determined with $\chi^2$-tests or Fisher's exact tests. Significant differences in other measures between dSPNs and iSPNs were determined with $t$-tests or Wilcoxon's rank-sum tests. Behavioral measurements associated with the three odor cues were compared with the one-way ANOVA followed by post-hoc Tukey's test. All statistical tests were two-tailed and $p$ values <0.05 were considered significant unless otherwise noted. All data are expressed as means±SEM unless otherwise noted.

**Data availability**. The data that support the findings of this study are available from the corresponding author upon reasonable request.

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

## Acknowledgements

We thank Daeyeol Lee for his helpful comments on the initial manuscript. This work was supported by the Research Center Program of Institute for Basic Science (IBS-R002-G1) (to M.W.J.) and National Research Foundation Grant (NRF-2016-Fostering Core Leaders of the Future Basic Science Program/Global Ph.D. Fellowship Program) (to J.H.S.).

## Author contributions

J.H.S and M.W.J. designed the study, analyzed the data, and wrote the manuscript. J.H.S. and D.K. performed the experiments.

## Additional information

**Competing interests:** The authors declare no competing financial interests.

