## [Peer Review File · Nature Communications]

Reviewers' comments:

Reviewer #1 (Remarks to the Author):

Here the authors perform a challenging set of experiments in head fixed mice: record from identified (otopogenetically-tagged) populations of MSNs in the dorsal medial striatum of mice performing a conditioned behavior. The task design is nicely designed to reveal distinct neural correlates associated with expected reward and unexpected reward omission nicely parameterized by the probability of reward being differential across 3 readily distinguishable olfactory cues. The authors provide behavioral evidence that animals have learnt the difference between the cues. Likewise the authors should be commended for attempting a quite detailed analysis by examining the correlates of each recorded cell across a variety of covariates using a fairly extensive regression model. Despite these merits of the task design, the recordings and analysis, the manuscript lacks a particularly strong conclusion and adds another set of observations to a relatively complex and confusing set of results in the field (that are relatively well summarized in the introduction to the paper). The introduction seems to suggest the problem. For example, near the end of the introduction the authors suggest "It is possible the direct pathway processes reinforcement-related signals and the indirect pathway processes punishment-related signals. Alternatively, both pathways may process similar signals but mediate different aspects of reward-based learning via differences in their output connectivity." I struggle to follow how the experiments in this paper are designed to shed light on this question. The only distinction above is with punishment per se (as opposed to absence of reward) otherwise the two alternatives are not strictly alternatives.

This led me to attempt to evaluate the paper more from the perspective of whether this is a particularly informative dataset useful to theorize about or design future experiments since from my perspective it adds another dataset of subtle differences between direct and indirect pathway neurons in the context of execution of a learned Pavlovian task. It does not, however, provide a clear framework with which to interpret or further our understanding of "the ways the direct and indirect pathways of the basal ganglia contribute to reward-based learning". In general, I do find it to be a high quality dataset that provides a good view of the diverse correlates in MSNs. As far as I can tell all of these correlates have been observed previously in various studies, but this provides a relatively comprehensive account that has value. Nonetheless, as the authors point out in the introduction, there is not a lack of papers describing subtle and hard to interpret differences between D1+ and D2+ MSNs.

My general concern about the value of the dataset is that there was a lack of analysis separately of the two main mouse lines used in these experiments - D1-cre and D2-cre - yet the main contrasts reported are across these two different cohorts of mice. Since, as the authors show, MSN activity is correlated with behavior (licking) it is important to examine how or whether the two cohorts of animals differ in their behavior. Since + identified D1 neurons were only in D1-cre mice and the converse for D2-cre mice the extent to which a reader may be confident that the difference across cell types is independent of differences in the behavioral expression of learning across the mouse lines. As the authors know there

have been many suggestions in the literature that the two mouse lines are somewhat different in their behavior, although careful quantification has indicated that some differences are negligible. I was particularly concerned because in some cases the differences are pretty subtle differences in the time course over which neurons were significantly modulated suggesting that behavioral differences could be driving dissociations. For example, Figure 3 middle row and Figure 4 third row.

Specific concerns:

I had difficulty understanding whether the authors eliminated putative TANs tagged in the D2-cre mice from the identified indirect pathway neurons used in analysis. They seem to have done some nice procedures to control for the concern that the D2-cre mice is known to also express cre in cholinergic interneurons (thought to be TANs), however, in the methods it is not completely clear whether data reported on indirect pathway neurons was done with and without including this population. Since a number of D1/D2 MSN differences turn on small percentages of the population (order or 10%) it is important to ensure this is not due to the inclusion of a difference population of non-indirect pathway neurons. I think the authors are saying they did do this here "Most of the optogenetically confirmed neurons (77 of 94 in D1-Cre mice; 75 of 102 in D2-Cre mice) were putative MSNs, which we then subjected to further analysis (Fig. S2 shows sample activity patterns from the different striatal neuron types; Fig. S3 shows the responses of all optogenetically confirmed D1 and D2 MSNs to laser stimulation)" but the methods section made me less clear about whether this was the case or not.

For the tagging experiments I was unclear whether the authors also considered waveform similarity as has previously been done (e.g. Cohen et al)? Some of the waveforms of positively tagged neurons appear quite divergent for light evoked versus non-light evoked (S3A first column last row, second row last column, first row second to last column). It would be valuable to reanalyze some key differences with a more stringent criterion if not.

I was quite impressed with this control: "As a control for odordependent, rather than value-dependent neuronal firing, we examined the effect of reversing cue-reward probability relationship on value-dependent striatal neuronal activity in a separate group of mice (n = 3). A significantly larger population showed similar than reversed activity relationships with value before and after the reversal, indicating that MSN responses to reward value cannot be accounted for by sensory responses to odor cues (Fig. S4A-C)." It is rarely done, but is a very nice control experiment.

Reviewer #2 (Remarks to the Author):

Summary: The authors performed an ambitious set of recording studies that aimed to elucidate the different roles of direct and indirect pathway medium spiny neurons in encoding and learning about reward paired cues and receipt of rewards. While the experiments were interesting, several methodological concerns made it difficult for us to support the conclusions as they are presented.

1. This study relied on optogenetic "tagging" of direct and indirect pathway neurons. Optogenetic tagging is difficult and can be prone to mis-interpretation if not done stringently. Most problematic for this study, I worry that their "tagged" neurons include multi-units (more than one neuron contributing spikes to the "tagged" unit). They authors note that approximately 1/4 of their "tagged" neurons had waveforms that were not consistent with medium spiny neurons. While they exclude these "tagged" units from further analysis, I worry that the methods they used for tagging medium spiny neurons were not stringent enough. In figure S1 the TAN and FSI clusters each contain "tagged" neurons from both D1 and D2-cre mice. As ChAT neurons don't express D1 receptors, and PV neurons don't express D2 receptors, this reinforces my worry that their tagging was not stringent enough, and that their analyses were contaminated by spikes from other striatal cell types. This criticism colored my evaluation of the rest of their study, as their "tagged" subpopulations in the D1 and D2 cre lines may have been very similar with each containing mixed populations of spikes from direct pathway neurons, indirect pathway neurons, and interneurons.

On a related point, the authors should have provided more quantitative information on the quality of their recordings. They should report the amplitude of recorded neurons vs the noise band, state how many recordings were performed in each mouse, and if the same mice from Figure 1 were the mice that were recorded from in figure 2.

2. The authors modeled firing rate using value, reward, and lick rate during three analysis windows (cue, delay, and reward). The use of the word value ($V(t)$) to signify reward probability is not correct, as the authors altered probability, and not value. Additionally, the authors should clarify their choice of regression analysis. What evidence is there to support the inclusion of all components in the model, especially previous trial information? Did the authors also look at the fit of other regression models, by perhaps omitting one component per window- lick rate in the cue window or reward probability in the reward window? If there is some basis for their inclusion, it seems that the components should be adjusted based on the trial window under analysis. When identifying value information in the cue window, why are trial outcome ($O(t)$) and lick rate ($L(t)$) included? Perhaps only the outcome of the previous trial ($O(t-1)$) should be included?

3. The light stimulus at the at the end of the delay period was only presented during the unrewarded trials, which produced a sensory confound in that the mouse could simply wait to see if a light turned on to learn the outcome of the trial. As such, it is difficult to disentangle the "reward-related signal" observed during the unrewarded trials from the sensory response to the light. The supplemental control experiment (Figure S4) is not comprehensive enough to address this concern. The experiment does not seem to have a behavioral component (ie. the light doesn't signal availability of reward) nor does it map on to the pavlovian task in terms of the temporal windows chosen for comparison. Finally, even with these caveats, many neural responses actually tracked the light cue and not the reward (blue dots in S4 panel B).

4. The analyses of reward prediction error were not convincing. The authors operationally defined reward prediction error, as a change in firing rate "during the reward period as a

function of reward value in the reward, but not unrewarded trials". I don't consider this reward prediction error, and am not sure how the authors came up with this. Reward prediction error is typically defined as the introduction of unexpected rewards, or the omission of expected rewards. As their experiments don't include any explicit behavioral probes for reward prediction error, I think this analysis and discussion should be removed.

Reviewer #3 (Remarks to the Author):

In this paper, the authors recorded electrical activity of Direct and Indirect medium spiny neurons during a probabilistic Pavlovian conditioning task. ChR2 photostimulation was used to identify D1 and D2 receptor-expressing MSNs, which belong to direct and indirect pathway MSNs, respectively. The main conclusion drawn from this paper is that at both direct and indirect pathway MSNs display increased activities related to reward value. When they separated the populations according to their changes of activity, they found that when CS values were increased, a bigger fraction of neurons that showed incremental increase in their activity belongs to D1 MSNs. On the other hand, in the population of neurons that decreased their activity, the proportion of D2 MSNs was higher. When they compared the neuronal responses to rewarded and no rewarded trials they found that both populations of MSNs increased their activity in similar ways when the reward was presented, however, immediately after the reward is presented, a larger portion of D2 neurons negatively correlated their activity to the reward. Furthermore, using PCA they were able to identify a particular dynamic where majority of neurons that were activated by no reward and/or inhibited by reward were mostly D2 MSNs, probably responsible of the negatively correlation of D2s to rewards.

The authors successfully described direct and indirect pathway activity on this particular conditioning task. During the reward related analysis, they carefully manipulated the variables to demonstrate that the neuronal activity is actually related to CV or rewards, but not by the other causalities (e.g. Odor identity or nature of no-reward) and they divided the task to subtract these information from every section. Overall, the results presented in this work are high quality and are helpful for supporting the idea that the participation of both striatal pathways are necessary for reward-based learning and nicely complements their previous works. On the other hand, the results obtained from motor behavior were not challenged by the authors in anyway as they did with the reward's data, demonstrating similar results previously published by other groups (e.g. Jin et al., 2014). The current manuscript can be improved to clarify a few concerns.

1. This study focuses on examining the activity of dorsomedial striatum. I think it is helpful to clearly state this in the title, and abstract, as the authors did not further compare the neural activity of direct/indirect MSNs in dorsolateral and/or ventral striatum.
2. The recent literature describing direct and indirect pathway MSNs often use mixed nomenclature, such as direct/indirect pathways, D1- D2 MSNs, etc. These terms can be confusing for general neuroscience audience who are not basal ganglia expert. One suggestion, to use dSPN and iSPN (direct pathway spiny projection neurons vs indirect pathway spiny projection neurons) throughout the paper. This way, one term is helpful for

defining both pathway, and cell type.

3. Separate your data in D1 and D2 MSNs when you challenge light neuronal response on supplementary figure 4D, to be consistent with your data. It is important to demonstrate that this effect is not happening in D2 MSNs.

4. It has been shown that manipulation of direct and indirect pathways could dis-engage animals from performing sequential motor behavior (Tecuapetla et al., 2015, 2016). Your conclusion could be strengthened by demonstrating that your recordings are indeed related to this motor behavior, then optogenetic stimulation should be able to at least modify start, stop, or delay licking bouts.

5. What is the proportion of neurons that participate in more than one stage of the task? Both responsive to CS and licking? Can you treat the data as a continuous instead of independent sections.

6. In figure 5, the authors showed that at least 4 types of responses. What is the proportion of D1 and D2 MSNs in type 4? In addition, I think is important to show which proportion of the total population of MSNs belong to each type of activity. Authors claim that type 3 neurons are the ones responsible for the different of negative correlate activity observed in 4B-C. If this statement is true, authors should be able to see the opposite correlation of an increment of D2 MSNs during the first 0.5s after no-reward trials.

7. In figure 3, the D2 MSNs look like having earlier onset in activity, is this statistically significant?

Response to reviewers' comments

Authors: Shin, Kim & Jung

Manuscript NCOMMS-17-07478, "Differential coding of reward and movement information in the dorsomedial striatal direct and indirect pathways"

We are grateful to the reviewers for their positive comments and constructive suggestions. In this revised manuscript, all of their suggestions have been incorporated as summarized below.

<Reviewer #1>

(1) Here the authors perform a challenging set of experiments in head fixed mice: record from identified (optogenetically-tagged) populations of MSNs in the dorsal medial striatum of mice performing a conditioned behavior. The task design is nicely designed to reveal distinct neural correlates associated with expected reward and unexpected reward omission nicely parameterized by the probability of reward being differential across 3 readily distinguishable olfactory cues. The authors provide behavioral evidence that animals have learnt the difference between the cues. Likewise the authors should be commended for attempting a quite detailed analysis by examining the correlates of each recorded cell across a variety of covariates using a fairly extensive regression model. Despite these merits of the task design, the recordings and analysis, the manuscript lacks a particularly strong conclusion and adds another set of observations to a relatively complex and confusing set of results in the field (that are relatively well summarized in the introduction to the paper). The introduction seems to suggest the problem. For example, near the end of the introduction the authors suggest "It is possible the direct pathway processes reinforcement-related signals and the indirect pathway processes punishment-related signals. Alternatively, both pathways may process similar signals but mediate different aspects of reward-based learning via differences in their output connectivity." I struggle to follow how the experiments in this paper are designed to shed light on this question. The only distinction above is with punishment per se (as opposed to absence of reward) otherwise the two alternatives are not strictly alternatives.

Response: We apologize for causing an unnecessary confusion on this matter. It seems that our description of the problem erroneously gives an impression that we tried to test which of the two alternatives is correct. Our intention was to mention two extremes that can explain effects of D1+ and D2+ neuronal stimulation on the animal's behavior. We now modified the introduction as the following (P 3, L 6 from bottom): "Currently, the ways the direct and indirect pathways of the basal ganglia contribute to reward-based learning remain poorly understood. The direct and indirect pathways may mediate behavioral promotion and suppression, respectively, based on different types of signal processing and/or different output connectivity. At one extreme, the direct and indirect pathways may exclusively process reinforcement- and punishment-related signals, respectively, with the same output connectivity. At the other extreme, both pathways may process similar signals but mediate different aspects of reward-based learning via differences in their output connectivity. And between these extremes, the two pathways may process quantitatively different reinforcement- and punishment-related signals with overlapping, but different output connectivity."

(2) This led me to attempt to evaluate the paper more from the perspective of whether this is a particularly informative dataset useful to theorize about or design future experiments since from my perspective it adds another dataset of subtle differences between direct and indirect pathway neurons in the context of execution of a learned Pavlovian task. It does not, however, provide a clear framework with which to interpret or further our understanding of "the ways the direct and indirect pathways of the basal ganglia contribute to reward-based learning". In general, I do find it to be a high quality dataset that provides a good view of the diverse correlates in MSNs. As far as I can tell all of these correlates have been observed previously in various studies, but this provides a relatively comprehensive account that has value. Nonetheless, as the authors point out in the introduction, there is not a lack of papers describing subtle and hard to interpret differences between D1+ and D2+ MSNs.

Response: Even though there are preceding studies related to ours, we'd like to point out that our study contains several novel findings. First, our results on D1+ and D2+ MSN activity related to value, reward, and reward prediction error are totally new. Previous physiological studies on neural correlates of D1+ and D2+

MSNs are all about movement. Also, previous studies on reward-based learning are all modulation studies (pharmacological or optogenetic manipulation of D1+ versus D2+ MSNs). Thus, it has been unclear *how* D1+ and D2+ MSNs process value- and reward-related information. In this respect, we believe that our results provide new insights into striatal neural processes underlying value and reward processing. For example, that value is represented by relative activity levels between D1+ and D2+ MSNs helps us to understand how the direct and indirect pathways work together in controlling behavior. Second, our finding that D1+ MSN activity is strongly correlated with movement offset (lick offset) is totally new. It not only calls for rethinking of the traditional dichotomous model, but also suggests a new role of the direct pathway in suppressing on-going behavior. Finally, we'd like to point out that the majority of previous studies targeted the dorsolateral striatum, whereas our study is on the dorsomedial striatum. Given that DMS and DLS have distinct anatomical connectivity and serve distinct roles, our findings will help understand how different cortico-striatal loops process reward and movement signals.

(3) My general concern about the value of the dataset is that there was a lack of analysis separately of the two main mouse lines used in these experiments - D1-cre and D2-cre - yet the main contrasts reported are across these two different cohorts of mice. Since, as the authors show, MSN activity is correlated with behavior (licking) it is important to examine how or whether the two cohorts of animals differ in their behavior. Since + identified D1 neurons were only in D1-cre mice and the converse for D2-cre mice the extent to which a reader may be confident that the difference across cell types is independent of differences in the behavioral expression of learning across the mouse lines. As the authors know there have been many suggestions in the literature that the two mouse lines are somewhat different in their behavior, although careful quantification has indicated that some differences are negligible. I was particularly concerned because in some cases the differences are pretty subtle differences in the time course over which neurons were significantly modulated suggesting that behavioral differences could be driving dissociations. For example, Figure 3 middle row and Figure 4 third row.

Response: We agree that it is important to examine licking behavior of D1-Cre and D2-Cre mice separately. As suggested, we analyzed licking behavior of D1-Cre

and D2-Cre mice and found similar licking behavior between the two types of mice. These results are now shown in supplemental Fig. S1, i and j.

Specific concerns:

(4) I had difficulty understanding whether the authors eliminated putative TANs tagged in the D2-cre mice from the identified indirect pathway neurons used in analysis. They seem to have done some nice procedures to control for the concern that the D2-cre mice is known to also express cre in cholinergic interneurons (thought to be TANs), however, in the methods it is not completely clear whether data reported on indirect pathway neurons was done with and without including this population. Since a number of D1/D2 MSN differences turn on small percentages of the population (order or 10%) it is important to ensure this is not due to the inclusion of a difference population of non-indirect pathway neurons. I think the authors are saying they did do this here "Most of the optogenetically confirmed neurons (77 of 94 in D1-Cre mice; 75 of 102 in D2-Cre mice) were putative MSNs, which we then subjected to further analysis (Fig. S2 shows sample activity patterns from the different striatal neuron types; Fig. S3 shows the responses of all optogenetically confirmed D1 and D2 MSNs to laser stimulation)" but the methods section made me less clear about whether this was the case or not.

Response: Yes, we excluded putative TANs from the analysis. We made this clear in the revised Methods (P 16, L 1).

(5) For the tagging experiments I was unclear whether the authors also considered waveform similarity as has previously been done (e.g. Cohen et al)? Some of the waveforms of positively tagged neurons appear quite divergent for light evoked versus non-light evoked (S3A first column last row, second row last column, first row second to last column). It would be valuable to reanalyze some key differences with a more stringent criterion if not.

Response: Yes, only those units with spike waveform correlations between laser-driven and spontaneous spikes > 0.85 were qualified as optically-tagged neurons. We revised Methods to make this point clearer as the following (P 16, 2nd paragraph of the revised text): "To be qualified as laser-activated, neurons had to meet two criteria^{40,57}. First, the latency to the first spike during the 6 ms window

after laser stimulation onset should be significantly (log-rank test, $p < 0.01$) lower than the latency to the first spike in a similar 6 ms window in the absence of laser stimulation. Second, the correlations between laser-driven and spontaneous spike waveforms should be > 0.85 ." The three units the reviewer pointed out had spike waveform correlations of 0.891, 0.926 and 0.911, respectively. As suggested by the reviewer, we reanalyzed neural data with more stringent criteria for optical tagging (long-rank test, $p < 0.001$; spike waveform correlation > 0.95). The number of tagged MSNs decreased somewhat (D1, from 77 to 64; D2, from 75 and 72), but the results were essentially similar to the original ones (see the figure below that shows some important results). This is briefly mentioned in the revised text as the following (P 16, 2nd paragraph, last sentence): "Increasing the stringency of these criteria (i.e., log-rank test, $p < 0.001$; spike waveform correlation > 0.95) reduced the number of laser-responsive SPNs (dSPNs, from 77 to 64; iSPNs, from 75 and 72), but yielded similar results (data not shown)."

log rank $p < 0.001$ & wave form correlation > 0.95

Results obtained with more stringent criteria for optogenetic tagging ($p < 0.001$, log-rank test; spike waveform correlation > 0.95). The graphs show neural activity related to value (A, B; compare with Fig. 3b, c), reward (c, d; compare with Fig. 4b, c), previous reward (E, F; compare with Fig. 3e, f), reward prediction error (g; compare with Fig. 6d), lick onset (h; compare with Fig. 7e), and lick offset (i; compare with Fig. 7f).

(6) I was quite impressed with this control: "As a control for odor-dependent, rather than value-dependent neuronal firing, we examined the effect of reversing cue-reward probability relationship on value-dependent striatal neuronal activity in a separate group of mice (n = 3). A significantly larger population showed similar than reversed activity relationships with value before and after the reversal, indicating that MSN responses to reward value cannot be accounted for by sensory responses to odor cues (Fig. S4A-C)." It is rarely done, but is a very nice control experiment.

Response: Thank you for this positive comment.

<Reviewer #2>

Summary: The authors performed an ambitious set of recording studies that aimed to elucidate the different roles of direct and indirect pathway medium spiny neurons in encoding and learning about reward paired cues and receipt of rewards. While the experiments were interesting, several methodological concerns made it difficult for us to support the conclusions as they are presented.

1. This study relied on optogenetic "tagging" of direct and indirect pathway neurons. Optogenetic tagging is difficult and can be prone to mis-interpretation if not done stringently. Most problematic for this study, I worry that their "tagged" neurons include multi-units (more than one neuron contributing spikes to the "tagged" unit). They authors note that approximately 1/4 of their "tagged" neurons had waveforms that were not consistent with medium spiny neurons. While they exclude these "tagged" units from further analysis, I worry that the methods they used for tagging medium spiny neurons were not stringent enough. In figure S1 the TAN and FSI clusters each contain "tagged" neurons from both D1 and D2-cre mice. As ChAT neurons don't express D1 receptors, and PV neurons don't express D2 receptors, this reinforces my worry that their tagging was not stringent enough, and that their analyses were contaminated by spikes from other striatal cell types. This criticism colored my evaluation of the rest of their study, as their "tagged" subpopulations in the D1 and D2 cre lines may have been very similar with each containing mixed populations of spikes from direct pathway neurons, indirect pathway neurons, and

interneurons.

On a related point, the authors should have provided more quantitative information on the quality of their recordings. They should report the amplitude of recorded neurons vs the noise band, state how many recordings were performed in each mouse, and if the same mice from Figure 1 were the mice that were recorded from in figure 2.

Response: As the reviewer indicated, proper optogenetic tagging and unit isolation are fundamental issues for the validity of our conclusions. We also agree that we did not provide enough details about these issues in the original manuscript. We believe that the chance for our conclusions suffering from these issues is small for the following reasons:

Unit isolation: We are confident about our unit isolation for several reasons, even though potential spike contamination cannot be absolutely excluded without intracellular recording, which is a general issue for all extracellular recordings. First, the chance for two different neurons to be identified as a single neuron is extremely low with tetrode recording (similar spike waveforms on all four tetrode channels) unless small amplitude spikes (those neurons located sufficiently far away from all four tetrode channels) are included in the analysis. We included only those units with L-ratio < 0.1 and isolation distance > 19, which are relatively stringent criteria for unit isolation in the field (Schmitzer-Torbert et al., 2005., *Neuroscience*). Furthermore, similar results were obtained when we included only those MSN unit clusters with L-ratio < 0.05, which are very well-isolated unit clusters, in the analyses (72 D1 MSNs and 65 D2 MSNs; see the figure below). These results make low-quality spike isolation as an unlikely candidate for our results.

Results obtained with unit clusters with L-ratio < 0.05. The graphs show neural activity related to value (A, B; compare with Fig. 3b, c), reward (c, d; compare with Fig. 4b, c), previous reward (E, F; compare with Fig. 3e, f), reward prediction error (g; compare with Fig. 6d), lick onset (h; compare with Fig. 7e), and lick offset (i; compare with Fig. 7f).

Second, it is unlikely that spikes from FSIs or TANs (high rate neurons) contaminated MSN spikes because only low-firing units were classified as MSNs. For further confirmation, we performed the same analyses after excluding relatively high-rate MSNs (cut-off value, one SD below mean firing rate of TANs, 3.61 Hz; 68 D1 MSNs and 68 D2 MSNs were included in the analysis). As shown by the figure below, similar results were obtained.

Results obtained with MSNs with mean firing rates < 3.61 Hz. The same format as in the previous figure.

Third, 'spike contamination' between D1 and D2 MSNs would not weaken our findings about differences between D1 and D2 MSNs. Let's assume that there was spike contamination between D1 and D2 MSNs, although we think this is a very unlikely possibility (see above). This will reduce, rather than increase, the chance to find differences between D1 and D2 MSNs, because both groups will have mixed D1 and D2 MSN populations. One might then argue that spike contamination between D1 and D2 MSNs may have blurred differences between D1 and D2 MSN responses. Specifically, D1 and D2 MSNs may actually exclusively process positive and negative value signals, respectively, but we may have found only quantitative differences between D1 and D2 MSNs because of spike contamination between them. However, elevation of previous reward signals was exclusive to D2 MSNs; previous reward signals kept decreasing for D1 MSNs unlike those for D2 MSNs (Fig. 4e). Likewise, elevation of lick offset-related signals was largely exclusive to D1 MSNs (Fig. 7e, f). These results are unexpected if there was spike contamination between D1 and D2 MSNs to a degree to erroneously induce overlapping positive and negative value responses of D1 and D2 MSNs.

Optogenetic tagging: We are confident about our optogenetic tagging for the following reasons: First, we used relatively stringent criteria for optogenetic tagging. Only those units with significantly ($p < 0.01$, log-rank test) higher firing rates during the first 6 ms window after light stimulation were qualified as tagged neurons. In addition, their spike waveform correlations between stimulation and no-stimulation conditions had to be > 0.85 . These criteria are similar to or more stringent than those used in the majority of previous studies. For example, we used a wider time window (10 ms) and lower correlation cut-off value (0.8) in our previous study (Kim et al., 2016., *Neuron*). Second, we obtained similar results when we applied more stringent criteria ($p < 0.001$, log-rank test; spike waveform correlations > 0.95). Please see our response to comment #5 of the reviewer #1.

Regarding D1-tagged TANs and D2-tagged FSIs, we were aware of the previous studies that failed to find D1 receptor-expressing TANs (Bergson et al., 1995, *J Neurosci*), which made us pay special attention to unit isolation and classification issues. As the reviewer suggested, one possibility is that putative TANs and FSIs

were contaminated by spikes from other striatal cell types. We admit that it is more difficult to exclude this possibility than the possibility of MSN spike contamination by TAN or FSI spikes. However, because we excluded putative TANs and FSIs from the analysis, our conclusions are irrelevant to this issue.

Aside from the issue of the validity of our conclusions, we suspect that D1-tagged TANs and D2-tagged FSIs might represent genuine striatal cell types for the following reasons: First, putative MSN, TAN and FSI unit clusters were reasonably well segregated (Fig. S2). Moreover, we were somewhat conservative in unit classification, excluding 106 neurons as unclassified. Combined with our use of rather stringent unit isolation criteria (see above), these argue against the possibility that spike contamination is behind our finding of D1-tagged TANs and D2-tagged FSIs. Second, D1-tagged TANs were consistently and reliably identified. We identified D1-tagged TANs (n=10) from all five mice (n=5) and eight of them survived as D1-tagged TANs even when we used more stringent tagging criteria (log-rank test, $p < 0.001$; spike waveform correlation > 0.95). Third, although striatal cholinergic interneurons predominantly express D2 and D5 receptor mRNA, they also express D1 receptor mRNA albeit at a lower level (20-25%) (Yan et al., 1997, *J Neurophysiol*; Kawaguchi et al., 1995, *Trends Neurosci*). Thus, it may be that previous studies failed to find D1 receptor-expressing TANs because they are rare. Fourth, although the reviewer raised a concern about our finding of both D1- and D2-tagged FSIs, PV and somatostatin-expressing interneurons are known to express D2 and D1 receptors, respectively (Kawaguchi et al., 1995, *Trends Neurosci*). Because both PV and somatostatin neurons discharge at high rates (Koos and Tepper, 1999, *Nat Neurosci*, Beatty et al., 2012, *J Neurophysiol*), our FSIs are likely to consist of both cell types.

As suggested, we now provide detailed information about unit isolation (and other information suggested by the reviewer) in the revised manuscript; we indicate details of unit isolation (Methods, P 15, last paragraph and Fig. S2e) and how many units were recorded from each animal in Results (P 5, L 6 and L 15). We also revised the text to indicate the mice in Fig. 1 are the same as the mice in Fig. 2 (P 4, 1st and 2nd paragraphs of the results section).

2. *The authors modeled firing rate using value, reward, and lick rate during three analysis windows (cue, delay, and reward). The use of the word value (V(t) to signify reward probability is not correct, as the authors altered probability, and not*

value. Additionally, the authors should clarify their choice of regression analysis. What evidence is there to support the inclusion of all components in the model, especially previous trial information? Did the authors also look at the fit of other regression models, by perhaps omitting one component per window- lick rate in the cue window or reward probability in the reward window? If there is some basis for their inclusion, it seems that the components should be adjusted based on the trial window under analysis. When identifying value information in the cue window, why are trial outcome ($O(t)$) and lick rate ($L(t)$) included? Perhaps only the outcome of the previous trial ($O(t-1)$) should be included?

Response: As the reviewer indicated, trial outcome ($O(t)$) is irrelevant to neural activity during the cue and delay periods (before trial outcome is revealed). However, adding an irrelevant variable won't affect regression results much as long as the number of trials is sufficiently large as in our study (> 300 trials per session; i.e., statistical power is sufficiently strong). As shown in the figure below, similar results are obtained regardless $O(t)$ is omitted or not during the cue and delay periods. We used the regression model including $O(t)$ for the sake of simplicity (one model applied to all epochs) and to be consistent with our previous studies (e.g., Kim et al., 2009, 2013, *J Neurosci*). As we understand, it is a common practice in the field to use a model with a trial outcome term to analyze neural activity before and after trial outcome. Of course, it would be perfectly OK to use a model omitting $O(t)$ for the cue and delay periods. If the reviewer and editor strongly prefer this way of analysis, we're willing to replace the current results with those obtained with the model without $O(t)$.

Value-related neural signals (fraction of value-responsive neurons) determined with regression models with and without trial outcome ($O(t)$) term. Note similarity between the two before trial outcome (up to 2 s). The same format as in Fig. 3.

We included lick rate in the regression model for the analysis of neural activity during the cue and delay periods because the animals showed substantial licking responses during these time periods (Fig. 1). Regarding previous outcome and previous value terms, our previous studies in the rat striatum (and also other areas such as the prefrontal cortex and hippocampus) have shown that choice- and outcome-related neural activity persist until the next trial (Kim et al., 2013, 2009, *J Neurosci*). In line with these findings, our preliminary analysis indicated that both outcome- and value-related neural activity persist until the next trial in our study (see the figure below). We included both previous outcome and previous value terms to control for effects of these variables on neural activity. Please note that this is a conservative way of determining neural correlates. The chance for false positive would decrease by including independent variables that significantly affect neural discharge rates. Omitting a relevant variable may result in a spuriously determined neural correlate especially when the variable of interest is correlated with the omitted variable.

Neural signals related to lick rate (left), previous outcome (middle) or previous value (right). The shading indicates chance level (binomial test, $\alpha = 0.05$). The analysis was based on the following regression model (eq. 1): $F(t) = a_0 + a_1 \cdot O(t) + a_2 \cdot V(t) + a_3 \cdot L(t) + a_4 \cdot O(t - 1) + a_5 \cdot V(t - 1)$. As shown, D1 and D2 MSN activity was significantly related to all three variables.

3. The light stimulus at the end of the delay period was only presented during the unrewarded trials, which produced a sensory confound in that the mouse could simply wait to see if a light turned on to learn the outcome of the trial. As such, it is difficult to disentangle the “reward-related signal” observed during the unrewarded trials from the sensory response to the light. The supplemental control experiment (Figure S4) is not comprehensive enough to address this concern. The experiment does not seem to have a behavioral component (ie. the light doesn’t signal availability of reward) nor does it map on to the pavlovian task in terms of the temporal windows chosen for comparison. Finally, even with these caveats, many neural responses actually tracked the light cue and not the reward (blue dots in S4 panel B).

Response: We agree that our conclusions can be strengthened by performing an additional control experiment. In a new control experiment, we delivered the light cue not only at the onset of trial outcome in reward omission trials, but also after trial outcome in all trials. Thus, the control light stimulus was delivered in the context of Pavlovian conditioning task, but with no contingency with trial outcome. As shown in Fig. S4e-h, the control light stimulus induced no detectable changes in MSN activity. The results further indicate that negative outcome-related MSN responses cannot be accounted for by sensory cue-related responses. In our previous control experiments (daily reversal and daily extinction), some sensory-related responses were observed (Fig. S4d). In the new control experiment,

however, the control light stimulus induced no detectable changes in MSN activity. These results suggest that weak sensory cue-related responses found in the previous control experiments are likely because of incomplete reversal/extinction. We expect that they will decrease further with extended trials after reversal/extinction.

4. The analyses of reward prediction error were not convincing. The authors operationally defined reward prediction error, as a change in firing rate "during the reward period as a function of reward value in the reward, but not unrewarded trials". I don't consider this reward prediction error, and am not sure how the authors came up with this. Reward prediction error is typically defined as the introduction of unexpected rewards, or the omission of expected rewards. As their experiments don't include any explicit behavioral probes for reward prediction error, I think this analysis and discussion should be removed.

Response: We apologize for not clearly explaining this matter in the original manuscript. Of those neurons that were significantly responsive to reward (analyzed using all trials, eq. 1), value-responsive neurons in rewarded (or unrewarded) trials (eq. 2) with opposite response directions (e.g., firing rate increases with reward and decreases with value) were defined as positive (or negative) RPE-coding neurons. We revised the related text as the following to clarify this (P 7, 2nd paragraph, L 7; see also P 8, L 3): "This neuron fired more during rewarded than unrewarded trials when the trial outcome was revealed. It also showed reduced firing as a function of value in rewarded trials. This neuron, therefore, responded to both the actual outcome (reward) and the predicted outcome (value) in opposite response directions. Since RPE is the difference between actual and predicted outcomes, this is an example of an RPE-coding neuron for rewarded trials (i.e., positive RPE-coding neuron)."

<Reviewer #3>

1. *This study focuses on examining the activity of dorsomedial striatum. I think it is helpful to clearly state this in the title, and abstract, as the authors did not further compare the neural activity of direct/indirect MSNs in dorsolateral and/or*

ventral striatum.

Response: Done as suggested.

2. *The recent literature describing direct and indirect pathway MSNs often use mixed nomenclature, such as direct/indirect pathways, D1- D2 MSNs, etc. These terms can be confusing for general neuroscience audience who are not basal ganglia expert. One suggestion, to use dSPN and iSPN (direct pathway spiny projection neurons vs indirect pathway spiny projection neurons) throughout the paper. This way, one term is helpful for defining both pathway, and cell type.*

Response: Thank you for the suggestion. Done as suggested.

3. *Separate your data in D1 and D2 MSNs when you challenge light neuronal response on supplementary figure 4D, to be consistent with your data. It is important to demonstrate that this effect is not happening in D2 MSNs.*

Response: Even though it would be nice to examine optically-tagged D1 and D2 MSN responses to light in the control experiment (Fig. S4D), we examined untagged MSN responses to light in this particular experiment. Because light cue-related responses largely subsided, and because D1 and D2 MSNs are expected to be recorded with similar probabilities, the results suggest that negative outcome-related responses of D1 and D2 MSNs cannot be accounted for by pure sensory responses to the light stimulus. Note that we performed an additional control experiment in response to the comment #4 of the reviewer #2 (shown in Fig. S4E-H of the revised manuscript), and we did examine light responses of tagged D1 and D2 MSNs. The light stimulus induced no detectable changes in on-going activity of D1 MSNs, D2 MSNs, or untagged MSNs when it had no predictive value.

4. *It has been shown that manipulation of direct and indirect pathways could dis-engage animals from performing sequential motor behavior (Tecuapetla et al., 2015, 2016). Your conclusion could be strengthened by demonstrating that your recordings are indeed related to this motor behavior, then optogenetic stimulation should be able to at least modify start, stop, or delay licking bouts.*

Response: We performed the suggested experiment and found that light stimulation of D1 MSNs can suppress licking behavior. The results are shown in Fig. 7g-l of the revised manuscript.

5. *What is the proportion of neurons that participate in more than one stage of the task? Both responsive to CS and licking? Can you treat the data as a continuous instead of independent sections.*

Response: Many neurons were responsive to CS and/or licking across multiple stages. We now show a summary of neuronal responses in new Fig. S7.

6. *In figure 5, the authors showed that at least 4 types of responses. What is the proportion of D1 and D2 MSNs in type 4? In addition, I think it is important to show which proportion of the total population of MSNs belong to each type of activity. Authors claim that type 3 neurons are the ones responsible for the different of negative correlate activity observed in 4B-C. If this statement is true, authors should be able to see the opposite correlation of an increment of D2 MSNs during the first 0.5s after no-reward trials.*

Response: As suggested, we now show the proportion of D1 and D2 MSNs among type 4 neurons and the distribution of the total MSNs across the four types (revised Fig. 5). Regarding the relationship between type 3 neurons and early responses to no reward, 3 (4%) D1 MSNs and 10 (13%) D2 MSNs were type 3 neurons during the first 0.5 of the reward period, which deviated significantly from an equal distribution (Fisher's exact test, $p = 0.045$). These results are described in the revised text (P 7, 1st paragraph, L 6 from bottom). We also softened the statement as the following: "These results suggest that type 3 iSPNs contribute to the strong negative reward-coding signals in the early reward period ..." (P 7, 1st paragraph, L 7 from bottom).

7. *In figure 3, the D2 MSNs look like having earlier onset in activity, is this statistically significant?*

Response: Yes, the green triangles in Fig. 3 indicate significant differences between D1 and D2 MSNs. This is clearly indicated in Fig. 3 legend as the

following: “Green triangles indicate significant differences between dSPNs and iSPNs (χ^2 -test, $p < 0.05$).”

Reviewers' comments:

Reviewer #2 (Remarks to the Author):

The authors have updated their manuscript but unfortunately fail to address the main shortcomings of this paper. I tend to agree with Reviewer 1 that the paper lacks a strong conclusion and is largely a descriptive account of relatively minor differences between firing patterns of striatal projection neurons. Specifics that were not addressed to my satisfaction:

1) I remain unconvinced by the author's arguments on optical tagging. I still find it problematic that they detect so many interneurons with their tagging protocols in D1 and D2 cre mice. They raise the possibility that these neurons represent non-canonical D1R-expressing cholinergic neurons or D2R-expressing FSI-like neurons. This may be possible but this remains a speculation as they provide no direct evidence that their cre lines label these neuron types. They also rightly say that they can exclude interneuron-like waveforms from their SPN analyses, but excluding putative interneurons from analysis doesn't address the issue - I'm not so much concerned with whether they can reliably reject false positively tagged TAN and FSI-like neurons, I'm concerned that they won't be able to reject false positive iSPNs and dSPNs in recordings of the other type.

2) I'm still not in agreement with the author's definition of reward-prediction error coding. Reward prediction error is commonly tested by either omitting an expected reward or presenting an unexpected one, neither of which are done here.

Reviewer #3 (Remarks to the Author):

The authors have addressed my major concerns.

Responses to additional comments of reviewer #2

The authors have updated their manuscript but unfortunately fail to address the main short-comings of this paper. I tend to agree with Reviewer 1 that the paper lacks a strong conclusion and is largely a descriptive account of relatively minor differences between firing patterns of striatal projection neurons.

Response: As we indicated in our response to the comments by the reviewer #1, we'd like to emphasize that our manuscript contains several new findings that significantly advance our understanding of *how* the direct and indirect pathways work together in controlling behavior. For example, we found quantitatively different positive and negative value-related neural activity between the direct and indirect pathways. It is totally unknown how value information is represented in the direct and indirect pathways of the striatum, because no study so far has examined value-related activity of direct- and indirect-pathway neurons separately. The direct and indirect pathways may process similar value-related signals, but control reward-based learning antagonistically based on different output connectivity. Alternatively, they may represent value information antagonistically in the first place. Our results suggest that the striatum may signal estimated value through the relative activity levels between the two pathways. This scheme has not been proposed previously, and hence our results provide a new insight on how the direct and indirect pathways may work together in controlling reward-based learning. Recent physiological studies on the direct and indirect pathways focused on movement-related neural activity in the dorsolateral striatum, finding the classic rate model of movement control is no longer tenable. Our results echo this conclusion in that both pathways in the dorsomedial striatum contain activity-increasing as well as -decreasing neurons as a function of value. Note, however, that our study goes one step further. Whereas findings on movement control so far stopped short of proposing a plausible neural process underlying movement control, our results provide a mechanistic explanation of how the two pathways might work together in processing value information. Regarding movement control, we'd like to point out that strong activation of D1 neurons in association with lick offset was totally unexpected. We further found that optogenetic stimulation of D1 neurons suppresses licking behavior while increases random movement. These results do not merely show a slight difference between the direct and indirect

pathways, but suggest a novel perspective on striatal circuit operation underlying movement control, because the direct and indirect pathways have been thought to antagonize one another by facilitating and suppressing movement, respectively. In these respects, we believe that our findings significantly advance our understanding on striatal neural processes underlying reward-based learning and motor control.

1) I remain unconvinced by the author's arguments on optical tagging. I still find it problematic that they detect so many interneurons with their tagging protocols in D1 and D2 cre mice. They raise the possibility that these neurons represent non-canonical D1R expressing cholinergic neurons or D2R-expressing FSI-like neurons. This may be possible but this remains a speculation as they provide no direct evidence that their cre lines label these neuron types. They also rightly say that they can exclude interneuron-like waveforms from their SPN analyses, but excluding putative interneurons from analysis doesn't address the issue - I'm not so much concerned with whether they can reliably reject false positively tagged TAN and FSI-like neurons, I'm concerned that they won't be able to reject false positive iSPNs and dSPNs in recordings of the other type.

Response: We agree with the reviewer that it is uncertain why both TANs and FSIs were optogenetically tagged in both D1-Cre and D2-Cre mice. To address this issue, we performed immunohistochemistry using brain slices obtained from the D1-Cre and D2-Cre mice that were used in the main experiments. We quantified the fractions of ChR2-expressing neurons that also express choline acetyltransferase (ChaT) or parvalbumin. We found that ChaT and parvalbumin are co-expressed with ChR2 in both D1-Cre and D2-Cre mice. Subpopulations of ChaT-positive neurons co-expressed ChR2 (6.6 and 6.2% in D1-Cre and D2-Cre mice, respectively), and the co-expressions were found in all animals tested, which is consistent with our physiological results (detecting D1- and D2-tagged TANs in all animals tested). We also found that subpopulations of parvalbumin-positive neurons co-express ChR2 (1.2 and 2.5% in D1-Cre and D2-Cre mice, respectively). These results explain why both TANs and FSIs were optogenetically tagged in both D1-Cre and D2-Cre mice. These results are now shown in Fig. S3 of the revised manuscript. As elaborated in our previous rebuttal, we obtained similar results when we used new unit-isolation criteria (L-ratio < 0.05) and optogenetic-tagging criteria ($p < 0.001$, log-rank test; spike waveform correlations > 0.95) that are way

more stringent compared to those used in other studies. We also showed that elevated neural activity related to previous reward was exclusive to iSPNs (Fig. 4e) and elevated neural activity related to lick offset was exclusive to dSPNs (Fig. 7e-f), which are unexpected if there was a substantial level of spike contamination between dSPNs and iSPNs. Combined with the new immunohistochemistry results, we believe that these results constitute very strong evidence for the validity of our optogenetic tagging.

2) I'm still not in agreement with the author's definition of reward-prediction error coding. Reward prediction error is commonly tested by either omitting an expected reward or presenting an unexpected one, neither of which are done here.

Response: As the reviewer indicated, the initial demonstration that dopamine neural activity is consistent with RPE was based on omitting an expected reward or presenting an unexpected one (Schultz et al., 1997, Science). Please note, however, that such manipulations are insufficient to dissociate neural activity related to RPE from that related to valence. RPE is the *quantitative* difference between actual and predicted outcomes. This is a critical point because it allows adjustment of reward value in proportion to the magnitude of error in predicting an actual outcome. Valence, on the other hand, simply provides information on whether an outcome is good or bad. Neural activity that is consistent with RPE in unexpectedly rewarded or unrewarded trials might actually represent valence-related activity (better or worse than expected, but not *how much* better or worse). For this reason, since the study by Bayer and Glimcher (2005, Neuron), quantitative manipulations are increasingly used to study RPE-related neural activity. Two schemes are popular in particular. First, the magnitude of reward is varied. Second, the probability of reward is varied. Both schemes have been used widely, and the following are a few examples that studied RPE by manipulating expected reward probability as in our study: Fiorillo, Tobler & Schultz, 2003, Science; Kennerley, Behrens & Wallis, 2011, Nature Neurosci; Sul et al., 2011, Neuron.

We'd also like to point out that our manipulation is formally similar to the manipulations the reviewer mentioned. Such manipulations (omitting an expected reward or presenting an unexpected one) should be repeated multiple times for a sufficient sampling of neural activity because activity of a single neuron can be

noisy. In omitting an expected reward, the expected reward probability is 100% only for the first omission and becomes less than 100% from the second omission, however small it is. Likewise, in presenting an unexpected reward, the expected reward probability is larger than 0% from the second presentation. Thus, previous studies using such manipulations measured neural responses to a reward (outcome = 1) or no reward (outcome = 0) against a certain level of expected reward probability. The structure of our task is similar. We measured neural responses to a reward (outcome = 1) or no reward (outcome = 0) against three different levels of expected reward probability (20, 50 and 80%). Hence, both schemes allow to measure neural responses to the difference between actual and predicted outcomes (i.e., RPE), although the degree of surprise differs (close to 0.2, 0.5 and 0.8 in our study and close to 1 in the studies using the manipulations the reviewer mentioned; see the figure below). Note that our task contains multiple levels of RPE for both rewarded (positive RPE) and unrewarded (negative RPE) trials, allowing us to identify neural activity quantitatively representing RPE.

REVIEWERS' COMMENTS:

Reviewer #2 (Remarks to the Author):

The authors make strong, well reasoned, arguments against my first and third points and I am satisfied with their manuscript with respect to these.

I remain worried about why their optical tagging is picking up so many interneurons, and whether this reflects an underlying issue with their method that may play out in the MSN population as well. To recap, 18% of the tagged neurons in D1-cre and 26% of the neurons in D2-cre mice were classified as interneurons by waveform parameters. While they say that, "Most of the optogenetically confirmed neurons (77 of 94 in D1-Cre mice; 75 of 102 in D2-Cre mice) were putative SPNs", they gloss over that most of their recorded neurons were SPNs. As interneurons made up ~24% of their total recorded neurons, they were "tagged" at roughly the same rate as SPNs in these experiments. I raised this as a concern about the specificity of their tagging protocol, as the D1-cre is not known to label ChAT interneurons and the D2-cre is not known to label PV interneurons. The authors now perform immunostaining to show that ~6% of ChAT and ~1% of PV interneurons are in fact labeled in both lines. While this number is much lower than what they report as "tagged" in their in vivo experiments, I cannot rule out the possibility that these interneurons do in fact express ChR2 with their expression strategy. To me this immunostaining suggests some leak in their Cre targeting strategy, as ChAT interneurons do not express D1R and PVs do not express D2R.

This said, all papers have some technical limitations and I would not hold the paper up over this point if it were adequately discussed. The authors have currently buried this quantification in the supplemental figure and caption. The only comment they make in the text regarding this issue is, "Fig. S3 shows immunohistochemical results for striatal interneurons". No interpretation or rationale for this experiment is given, nor any discussion of this as a potential technical limitation. As the quality of the tagging is crucial to the interpretation of their findings on SPNs, they should describe this result and explain why they performed the immunostaining in the main text. In particular, they should be clear that they tagged interneurons at roughly the same rate as SPNs, and link it to expression of ChR2 in ChAT and PV interneurons of both cre lines.

Reviewer #4 (Remarks to the Author):

I've now reviewed the paper as well the rebuttal. I do share the three main concerns the referee #2 raised, including 1) the descriptive nature of D1/D2-SPNs activity in the study without a conceptual framework on how the two pathways might actually work; 2) the D1- and D2-cre lines the authors used are not clean enough to specifically target D1- and D2-SPNs but consist of a significant proportion of both ChAT and PV interneurons; 3) I also remains to be convinced by the reward-prediction error (RPE) conclusion, not from the RPE definition aspect as referee #2 pointed to but from the data itself. For instance, the example neuron shown in Fig. 6a exhibited dramatically different baseline firing rate before reward

onset under various probability tasks, suggesting that it might also encode reward expectation, uncertainty or simply licking movements.

The value encoding part (Fig. 3) of the current study seems new, but the movement control parts are not. Different from what the authors claimed in the rebuttal that "strong activation of D1 neurons in association with lick offset was totally unexpected", it has been well known that both D1 and D2 pathways are involved in action sequence termination and movement offset (see refs 13, 14 & 19).

Response to reviewer's comments

Reviewer #2

I remain worried about why their optical tagging is picking up so many interneurons, and whether this reflects an underlying issue with their method that may play out in the MSN population as well. To recap, 18% of the tagged neurons in D1-cre and 26% of the neurons in D2-cre mice were classified as interneurons by waveform parameters. While they say that, "Most of the optogenetically confirmed neurons (77 of 94 in D1-Cre mice; 75 of 102 in D2-Cre mice) were putative SPNs", they gloss over that most of their recorded neurons were SPNs. As interneurons made up ~24% of their total recorded neurons, they were "tagged" at roughly the same rate as SPNs in these experiments. I raised this as a concern about the specificity of their tagging protocol, as the D1-cre is not known to label ChAT interneurons and the D2-cre is not known to label PV interneurons. The authors now perform immunostaining to show that ~6% of ChAT and ~1% of PV interneurons are in fact labeled in both lines. While this number is much lower than what they report as "tagged" in their in vivo experiments, I cannot rule out the possibility that these interneurons do in fact express ChR2 with their expression strategy. To me this immunostaining suggests some leak in their Cre targeting strategy, as ChAT interneurons do not express D1R and PVs do not express D2R.

This said, all papers have some technical limitations and I would not hold the paper up over this point if it were adequately discussed. The authors have currently buried this quantification in the supplemental figure and caption. The only comment they make in the text regarding this issue is, "Fig. S3 shows immunohistochemical results for striatal interneurons". No interpretation or rationale for this experiment is given, nor any discussion of this as a potential technical limitation. As the quality of the tagging is crucial to the interpretation of their findings on SPNs, they should describe this result and explain why they performed the immunostaining in the main text. In particular, they should be clear that they tagged interneurons at roughly the same rate as SPNs, and link it to expression of ChR2 in ChAT and PV interneurons of both cre lines.

Response: We agree that the quality of optogenetic tagging is a crucial issue. As suggested, we describe and discuss the results related to optogenetic tagging of putative interneurons in the main text (Results, P 5, 1st paragraph, L 9-16 and 2nd

paragraph; Discussion, P 12, 2nd paragraph). In doing so, we explicitly mention similar rates of optogenetic tagging for putative SPNs and interneurons, explain why we performed immunostaining, and discuss a possibility of nonspecific tagging. We also show unit classification results, which were part of Supplemental Figure 2, in the main figure (Figure 2).

Reviewer #4

I've now reviewed the paper as well the rebuttal. I do share the three main concerns the referee #2 raised, including 1) the descriptive nature of D1/D2-SPNs activity in the study without a conceptual framework on how the two pathways might actually work;

Response: As the reviewer pointed out, our manuscript contains elaborate descriptions of neural activity related to expected reward, actual reward, and tongue movement rather than testing a specific hypothesis. Note that the type of work that drives the advancement of a field differs depending on the status of a field. Currently, empirical findings on the operation of the direct and indirect pathways are way insufficient to impose strong constraints on plausible models of striatal circuit operation. Critical empirical observations would be particularly useful at this stage, and we believe that our study is an example. Studies aiming to test specific hypotheses would be feasible only after specific models have been established. Also note that we do not stop at simply describing neural correlates, but propose how the two pathways might work together. For example, we propose that the direct and indirect pathways may determine the likelihood with which an animal will choose a particular target based on the relative activity of the two pathways. We also propose that the dorsomedial direct pathway may selectively participate in terminating on-going behavior. As such, although our work may be descriptive, it is closely tied to theoretical issues of striatal circuit operation.

2) the D1- and D2-cre lines the authors used are not clean enough to specifically target D1- and D2-SPNs but consist of a significant proportion of both ChAT and PV interneurons;

Response: As the reviewer pointed out, our immunostaining results indicate that Ch2R is expressed not only in SPNs, but in ChAT- and PV-expressing interneurons

in both D1-Cre and D2-Cre mice, which explains why we obtained optically-tagged interneurons in both mouse lines. We therefore agree with the reviewer's comment that our mouse lines may not be 'clean' to specifically target D1- and D2-SPNs, which is acknowledged and discussed in the revised text (Results, P 5, 1st paragraph, L 9-16 and 2nd paragraph; Discussion, P 12, 2nd paragraph). Please note, however, that we excluded putative interneurons from the analysis. Please also note that Ch2R expressions are well segregated between dSPNs and iSPNs as revealed by immunostaining (Fig. 2) and largely selective dSPN and iSPN response elevations to lick offset and previous reward, respectively (Fig. 4 and 7). Therefore, our conclusions are not undermined by the optogenetic tagging of interneurons in a serious manner.

3) I also remains to be convinced by the reward-prediction error (RPE) conclusion, not from the RPE definition aspect as referee #2 pointed to but from the data itself. For instance, the example neuron shown in Fig. 6a exhibited dramatically different baseline firing rate before reward onset under various probability tasks, suggesting that it might also encode reward expectation, uncertainty or simply licking movements.

Response: Whether neural activity seemingly related to a variable of interest is actually related to other confounding factors is a fundamental issue in neurophysiology. We acknowledged this in the revised text (P 7, L 1 from bottom). We also revised the text to better explain how we handled this problem (analyzing rewarded and unrewarded trials separately and including lick rate in the multiple regression analysis; P 8, L 1-4). Please note that value- or reward probability-related neural activity is a component of RPE-related neural activity rather than a confounding variable. This is because RPE, by definition, is the difference between reward (actual outcome) and value (expected outcome). Also, please note that cue-dependent 'baseline' firing before reward onset is quite expected because both dSPN and iSPN populations convey strong value signals throughout the cue, delay and outcome periods (Fig. 3 of the revised manuscript). In fact, such patterns of value- and reward-related neural activity are found in widespread cortical and subcortical structures including the striatum; value signals arise before trial outcome and they are combined with reward signals once outcome is revealed to compute RPE (reviewed in Lee et al., 2012, Ann Rev Neurosci). Although midbrain dopaminergic neurons show phasic RPE-related discharges (no value-dependent firing immediately before trial outcome) under certain conditions, putative

GABAergic neurons convey value-dependent persistent activity before trial outcome in the ventral tegmental area (Cohen & Uchida, 2012, Nature; Eshel et al., 2015, Nature). Thus, maintaining value signals before and after trial outcome and integrating this information with reward signals might be a general characteristic for RPE-computing brain structures. We revised the text to make this point clear (P 7, last paragraph).

Whether or not a given neuron shows value-dependent (or any other variable-dependent) activity before outcome is irrelevant to the issue of whether a given neuron conveys RPE information when trial outcome is revealed. A given neuron may play a role of maintaining value (and other) information as well as computing RPE (like cortical and striatal neurons); alternatively, a given neuron may play a role of computing RPE, but not involved in maintaining value information (like dopaminergic neurons). In both cases, the neuron may play a role of computing RPE, which can be examined by testing whether it conveys both value and outcome signals *during the outcome period* and comparing their relative response directions. As can be seen in the example neuron (Fig. 6a of the revised manuscript), value-related neural activity is not maintained statically, but changes dynamically before and after trial outcome (mentioned in P 8, L 11). For these reasons, we think that it would be awkward to mention the reviewer's comment (the example striatal neuron might also encode reward expectation, uncertainty or simply licking movements) in the text. It would be more appropriate to acknowledge this issue in a general way ("Value-dependent firing may be confounded by other factors such as lick rate.", P 8, L 3).

The value encoding part (Fig. 3) of the current study seems new, but the movement control parts are not. Different from what the authors claimed in the rebuttal that "strong activation of D1 neurons in association with lick offset was totally unexpected", it has been well known that both D1 and D2 pathways are involved in action sequence termination and movement offset (see refs 13, 14 & 19)

Response: As the reviewer indicated, previous studies have shown concurrent activation of both D1 and D2 pathways in association with action sequence termination and movement offset. A novel aspect of our finding is selectivity. Lick offset-associated response increase was largely selective to dSPNs (Fig. 7 of the revised manuscript). We revised the text to make this point clear (P 9, 2nd paragraph, L 5).